

# Characteristics of extratropical cyclones and precursors to windstorms in Northern Europe

Terhi K. Laurila[1], Hilppa Gregow[1], Joona Cornér[2], and Victoria A. Sinclair[2]

[1]Weather and Climate Change Impact Research, Finnish Meteorological Institute, Helsinki, Finland
[2]Institute for Atmospheric and Earth System Research / Physics, Faculty of Science, University of Helsinki, Helsinki, Finland
**Correspondence:** Terhi K. Laurila (terhi.laurila@fmi.fi)

**Abstract.** Extratropical cyclones play a major role in the atmospheric circulation, weather variability and can cause damage to society. Extratropical cyclones in Northern Europe, which is located at the end of the North Atlantic storm track, have been less studied than extratropical cyclones elsewhere. Our study investigates extratropical cyclones and windstorms in Northern Europe (which in this study covers Norway, Sweden, Finland, Estonia and parts of the Baltic, Norwegian and Barents Seas) by

analysing their characteristics, spatial and temporal evolution and precursors. We examine cold and warm seasons separately to determine seasonal differences. We track all extratropical cyclones in Northern Europe, create cyclone composites and use an ensemble sensitivity method to analyse the precursors. The ensemble sensitivity analysis is a novel method in cyclone studies where linear regression is used to statistically identify what variables possibly influence the subsequent evolution of extratropical cyclones. We investigate windstorm precursors for both the minimum mean sea level pressure (MSLP) and

for the maximum 10-m wind gusts. The annual number of extratropical cyclones and windstorms have a large inter-annual variability and no significant linear trends during 1980–2019. Windstorms originate and occur over the Barents and Norwegian Seas whereas weaker extratropical cyclones originate and occur over land areas in Northern Europe. During the windstorm evolution, the maximum wind gusts move from the warm sector to behind the cold front following the strongest pressure gradient. Windstorms in both seasons are located on the poleward side of the jet stream. The maximum wind gusts occur

nearly at the same time than the minimum MSLP occurs. The cold season windstorms have higher sensitivities and thus are potentially better predictable than warm season windstorms, and the minimum MSLP has higher sensitivities than the maximum wind gusts. Of the four examined precursors, both the minimum MSLP and the maximum wind gusts are the most sensitive to the 850-hPa potential temperature anomaly i.e. the temperature gradient. Hence, this parameter is likely important when predicting windstorms in Northern Europe.

# 1 Introduction

Extratropical cyclones are important phenomena to regulate the daily weather in mid-latitudes and to transport energy and moisture in the atmosphere. They are mainly driven by baroclinicity (Charney, 1947) which is characterised by a strong temperature difference between the poles and equator. They are associated with precipitation and strong winds which can lead to high impacts in society (e.g. Suursaar et al., 2006; Kufeoglu and Lehtonen, 2014; Holley, 2021; Tervo et al., 2021; Rantanen





25 et al., 2021), for example floods, power outages, and forest and property damage. Since extratropical cyclones play a crucial role in the atmospheric circulation, weather variability and as a cause for societal and economical impacts, the knowledge and understanding of extratropical cyclone climate and dynamics are essential.

 Extratropical cyclones and their tracks have been widely studied in the Northern Hemisphere (e.g. Hoskins and Hodges, 2002; Ulbrich et al., 2009; Hodges et al., 2011; Priestley et al., 2020). It is well known that the main storm track regions

30 in the Northern Hemisphere are located over the North Atlantic and the North Pacific, and secondary storm track regions over the Mediterranean and Siberia (e.g. Hoskins and Hodges, 2002). The North Atlantic storm track region starts east of the Rocky Mountains in North America and ends in Northern Europe (e.g. Hoskins and Hodges, 2002; Priestley et al., 2020). In this study, we consider Northern Europe as a region covering Norway, Sweden, Finland, Estonia and parts of the Baltic, Norwegian and Barents Seas. In general, extratropical cyclones at the end of all storm track regions are less studied. However,

35 the structure and evolution of such extratropical cyclones may differ from extratropical cyclones elsewhere. Schultz et al. (1998) noted that extratropical cyclones at the end of the North Atlantic storm track generally resemble more of the Norwegian cyclone model (Bjerknes, 1919; Bjerknes and Solberg, 1922) than the Shapiro–Keyser cyclone model (Shapiro and Keyser, 1990) which is reasonable since the Norwegian cyclone model was developed based on cyclones that occur in Northwestern Europe. Wang and Rogers (2001) found that there are many differences in the dynamical and thermal structure and evolution

40 of strong explosive cyclones in the Northeastern Atlantic compared to the Northwestern Atlantic. For example, the explosive cyclones in the Northeastern Atlantic have lower static stability but less environmental baroclinicity than explosive cyclones in the Northwestern Atlantic (Wang and Rogers, 2001).

 There are two main ways to identify extratropical cyclones: the Eulerian approach and the Lagrangian approach (Hoskins and Hodges, 2002). The Eulerian approach uses basic statistics such as the variance or standard deviation to a chosen meteorological

45 field (for example mean sea level pressure (MSLP) or 850-hPa relative vorticity) which is bandpass filtered with synoptic time scales, usually 2–6 days (Blackmon, 1976). While the Eulerian approach is straightforward and gives a general description of the storm track activity it does not provide information about individual extratropical cyclones and their characteristics. The Lagrangian approach is a feature tracking method where a localised minimum or maximum of a meteorological field (for example MSLP or 850-hPa relative vorticity) is identified and their location in time is tracked (e.g. Hodges, 1994). With

50 the Lagrangian approach it is possible to analyse cyclone specific characteristics such as the genesis and lifetime. Moreover, additional meteorological variables can be determined along the cyclone track to investigate how they vary during the cyclone evolution.

 The spatial and temporal evolution of extratropical cyclones can be analysed by producing cyclone composites. Cyclone compositing allows to statistically find key features of the structure and development of the chosen set of cyclones. Cyclone

55 composites have been used in extratropical cyclone research for example for creating an extratropical cyclone atlas for the 200 most intense historical extratropical cyclones in the North Atlantic (Dacre et al., 2012) and to investigate how moisture is transported into extratropical cyclones (Dacre et al., 2019). Commonly cyclone composites are used to study the structures in extratropical cyclones for example for precipitation and clouds (Field and Wood, 2007) and for air–sea turbulent heat fluxes





and heat and moisture content (Rudeva and Gulev, 2011). Cyclone composites have also been used to examine extratropical

cyclones and their structures in climate models (Catto et al., 2010) and in warming climate (Sinclair et al., 2020).

Since extratropical cyclones are responsible of the daily weather and their associated heavy rainfall and strong winds can cause damage, it is important that extratropical cyclones are accurately predicted. In addition to producing forecasts directly for the parameters of extratropical cyclone intensity (for example MSLP or wind gust), we can use other variables as proxies to predict the windstorm intensity. The use of proxies in forecasting is especially helpful regarding parameters that are small in

scale and therefore harder for numerical weather prediction models to accurately predict. For forecasting purposes, the proxies should be descriptive of the evolution of windstorms earlier than the most intense phase i.e. they should act as precursors to windstorms. One way to find precursors to windstorms is to use ensemble sensitivity analysis (which is described in detail in Sect. 3.4) which allows us to quantify how sensitive the subsequent windstorm intensity is to precursor fields. The ensemble sensitivity analysis method was examined by Ancell and Hakim (2007) who analysed the linear connection between perturbed

initial conditions and forecast metrics of the wintertime flow pattern on the west coast of North America. The ensemble sensitivity method has also been used to find precursors to intense Mediterranean cyclones (Garcies and Homar, 2009) and to extratropical cyclones in the west and east North Atlantic (Dacre et al., 2012). This method is a computationally simple way to identify sensitivities, but it has a limitation of assuming a linear relationship between the precursor and the characteristics of the windstorm being predicted. Nonetheless, the results can provide guidance for forecasters and indicate to which variables

the most attention should be paid. From an application point of view, the sensitivities can also be used to assess, for example, likely error propagation or future climate projections. Since the resulting sensitivity shows how much, for example, the windstorm intensity changes if the precursor changes, therefore a similar magnitude change in precursor caused by an error in the numerical weather model would result in a similar change in the windstorm intensity than the sensitivity results show. Likewise, we can estimate how warmer climates would change the windstorm intensity by applying the sensitivity results to

how the precursors are projected to change.

In this study, we aim to increase the knowledge of the extratropical cyclone and windstorm climate in Northern Europe which occurs at the end of the North Atlantic storm track. Although many extratropical cyclone studies focus only on the cold season or winter months (e.g. Hoskins and Hodges, 2002; Pinto et al., 2005), we investigate the results separately for the cold (October–March) and warm (April–September) seasons in order to analyse the seasonal differences. In addition to the

climatological properties as temporal frequencies and cyclone characteristics, we extend the study to examine what precursors are relevant in Northern Europe windstorms by using ensemble sensitivity analysis. These results may help forecasters to better predict and prepare for windstorms by increasing their understanding of what environments windstorms typically develop in.

The research questions we aim to answer in this paper are: 1) What are the annual and monthly frequencies of extratropical cyclones and windstorms in Northern Europe?, 2) Are there differences in cyclone characteristics between extratropical

cyclones and windstorms and between the cold and warm seasons?, 3) How the spatial and temporal structure of Northern Europe windstorms evolve?, and 4) What precursor has the strongest impact on the minimum MSLP and the maximum wind gust in Northern Europe windstorms? The remaining paper is structured as follows: first, we introduce the data in Sect. 2 and the methods in Sect. 3. The annual and monthly frequencies of the three cyclone classes (all extratropical cyclones, windstorms





and non-windstorms) are shown in Sect. 4 and different cyclone characteristics are analysed in Sect. 5. Section 6 examines the
spatial and temporal evolution of windstorms, and Sect. 7 investigates the precursors to windstorms. The final conclusions are
given in Sect. 8.

## 2   ERA5 reanalysis data

ERA5 is the newest atmospheric reanalysis produced by the European Centre for Medium Range Weather Forecasts (Hersbach
et al., 2020) and it is based on the Integrated Forecasting System (IFS, cycle 41r2). ERA5 has a horizontal resolution of 31 km
and vertically it has 137 levels from the surface up to approximately 80 km. ERA5 provides hourly analysis and forecast fields
and while conducting the analysis for this study, ERA5 covered years from 1979 onwards. Recently, there was a release of a
back extension making ERA5 available from 1950 onwards.

    In our study, ERA5 data is obtained every 3 hours from 1980–2019 (40 years). The variables we obtain are: MSLP, the max-
imum 10-m wind gust (maximum since previous post-processing), 850-hPa temperature, total column water vapour (TCWV),
300-hPa horizontal wind speed components, and 300-hPa potential vorticity (PV). We consider 10-m wind gusts rather than
10-m wind speeds since the impacts and damage regarding extratropical cyclones are usually caused by the strong wind gusts
(Gardiner et al., 2013).

## 3   Methods

### 3.1   Extratropical cyclone tracking

The cyclones are tracked with the TRACK algorithm (Hodges, 1994, 1995, 1999) and the tracking is based on 3 hourly MSLP
from ERA5. The MSLP field is smoothed to T63 resolution (around 180 km) before running TRACK to decrease the noise in
the field and in addition, wavenumbers smaller than 5 are removed in order to exclude the large planetary scale waves. Since
we want to investigate large-scale extratropical cyclones, we included only tracks that last at least 1 day and move at least 500
km during their lifetime in order to remove the short lived and stationary cyclones. After the tracks are created, we then found
the locations and values of the minimum MSLP of the cyclone centre at each time step of each track from the native, high
resolution (31 km) data. We also found, for each time step of the track, the maximum 10-m wind gust with a 6° radius because
we wanted to find the wind gusts associated with the extratropical cyclones. The results presented in Sect. 6 confirm that the
6° radius used to identified the associated wind gusts is an appropriate choice.

### 3.2   Extratropical cyclone classes

The classification of the extratropical cyclones is made based on their associated wind gusts. In this study, we include all
extratropical cyclones that have at least one point of their track located inside a box 55–75°N, 5–40°E (the box is shown in
Fig. 4) during 1980–2019. The wind gust distribution of all extratropical cyclones is created by using the maximum 10-m wind
gust within 6 degrees of the cyclone centre at each time step when the cyclone centre of the track is inside the Northern Europe





box. Thus, there can be more than one gust value per cyclone track and additionally, the location of the maximum wind gust
can be at maximum 6 degrees outside the box (Fig. A2). Figure 1 shows the wind gust distributions covering the whole year
(Fig. 1a), the cold season (Fig. 1b) and the warm season (Fig. 1c). All of the distributions are slightly positively skewed i.e.
they have more weight in the right tail. However, the mean and median values are relatively close to each other and the mean
values differ from 22.2 m s$^{-1}$ in the cold season (Fig. 1b) to 18.1 m s$^{-1}$ in the warm season (Fig. 1c). The 90$^{th}$ percentile of
the wind gust distribution covering the whole year is 27.2 m s$^{-1}$ (Fig. 1a). While the 90$^{th}$ percentile of wind gusts during the
cold (29.4 m s$^{-1}$) and warm (23.6 m s$^{-1}$) seasons differ from the whole year value, these are however relatively near each
other compared to the maximum values which differ from 51.6 m s$^{-1}$ in the cold season to 38.1 m s$^{-1}$ in the warm season.

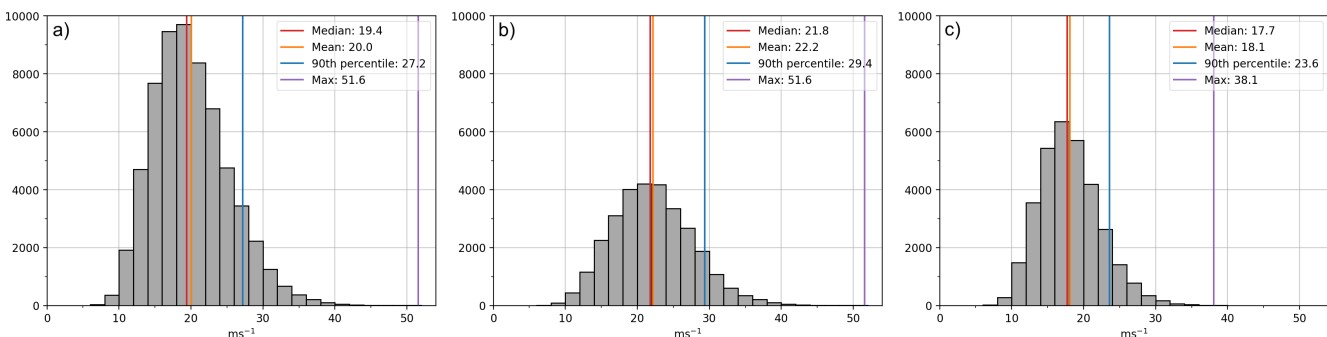

**Figure 1.** Distribution of the maximum 10-m wind gust within 6 degrees of the cyclone centre at each time step when the cyclone centre of
the track is inside the Northern Europe box during 1980–2019 covering a) the whole year, b) the cold season (October–March), and c) the
warm season (April–September).

In order to classify the extratropical cyclones based on their extremeness in wind gusts without depending on the season, we
use 27.2 m s$^{-1}$ (i.e. the 90$^{th}$ percentile of wind gusts during the whole year) as a threshold to divide the extratropical cyclones
to windstorms and non-windstorms. The choice of 90$^{th}$ percentile as a threshold gives a reasonable number of windstorms
for further analysis. When the windstorm threshold is equal in both seasons we can more easily compare the seasons since
otherwise the warm season windstorms would include extratropical cyclones that have lower wind gusts than cold season
windstorms. Therefore, our final cyclone classes are: 1) all extratropical cyclones; all extratropical cyclones that have at least
one track point inside the Northern Europe box, 2) windstorms; extratropical cyclones which associated wind gusts are equal to
or exceed the 90$^{th}$ percentile when the cyclone centre is inside the Northern Europe box, and 3) non-windstorms; extratropical
cyclones which associated wind gusts are below the 90$^{th}$ percentile when the cyclone centre is inside the Northern Europe box.
Hence, the sum of windstorms and non-windstorms equals all extratropical cyclones.

In this study, we also need to identify, for each windstorm, the time of the minimum MSLP and the time of the maximum
10-m wind gust. The time of the minimum MSLP is the time when the absolute minimum of MSLP occurs along the cyclone
track and therefore the cyclone centre does not need to be located within the Northern Europe box at this time. In contrast, as
we are interested in investigating wind gusts in Northern Europe, we define the time of the maximum 10-m wind gust to be the





time that the maximum gust occurs while the cyclone centre is within the Northern Europe box. However, as the gusts are taken within a 6° radius of the cyclone centre, the location of the maximum gust can occur just outside of the Northern Europe box. The locations of the minimum MSLP and the maximum 10-m wind gust for each windstorm track are shown in the Appendix Fig. A1 and Fig. A2.

**3.3 Cyclone composites**

To further examine the structure and evolution of Northern Europe windstorms (identified as described in Sect. 3.2), we create cyclone composites in similar way as for example Dacre et al. (2012) and Sinclair et al. (2020). The composites are produced for different meteorological variables at different offset times relative to two times: the time of the minimum MSLP and secondly the time of the maximum 10-m wind gust (these times are defined at the end of Sect. 3.2). For the compositing, the windstorms
are transformed into a spherical grid (cyclone centre in the middle) and rotated to a cyclone relative coordinate so that they all travel to the east. The composite for each meteorological variable and offset time is created by averaging that meteorological field of each individual windstorm.

**3.4 Ensemble sensitivity analysis**

To statistically identify precursors which are strongly correlated with characteristics of windstorms in Northern Europe, we use
an ensemble sensitivity analysis. In this study, "ensemble" refers to the population of extratropical cyclones that are defined as windstorms. The principle of the ensemble sensitivity method is to calculate the linear regression between a precursor field, $x$, and the so called response function, $J$. In our study, we consider two response functions: 1) the minimum MSLP, and 2) the maximum 10-m wind gust. The minimum MSLP was chosen because it is a common variable to represent the intensity of an extratropical cyclone, and the maximum wind gust was chosen because gusts usually cause the high impacts and damage. The
minimum MSLP is not directly related to the wind gusts but indirectly by strengthening the pressure gradient which leads to stronger wind speeds and hence stronger wind gusts. The ensemble sensitivity analysis is applied to precursors on the cyclone centered radial grid.

The sensitivity $S$ consists of multiplying three components:

$$S_{i,j} = m_{i,j}\alpha_{i,j}\sigma_{i,j} \tag{1}$$

where $m$ is the linear regression slope, $\alpha$ is a correction factor, and $\sigma$ is the standard deviation of the precursor. The linear regression slope $m$ is calculated at each grid point $(i,j)$:

$$m_{i,j} = \left(\frac{\delta J}{\delta x}\right)_{i,j}. \tag{2}$$





The correlation factor $\alpha$ modifies the sensitivity $S$ by giving less weight to slope values in those grid points that have weak correlations. This is done by defining $\alpha$:

$$\alpha_{i,j} = \begin{cases} 1 & \text{if } r_{i,j}^2 \geq r_{min}^2, \\ \frac{r_{i,j}^2}{r_{min}^2} & \text{if } r_{i,j}^2 < r_{min}^2 \end{cases} \qquad (3)$$


where $r_{i,j}$ is the correlation coefficient and $r_{min}$ is the threshold for correlations that are altered. Similarly to Dacre and Gray (2013), we set $r_{min}^2$ to be 0.05 which means that for all grid points where the correlation coefficient is less than 0.224, the sensitivity $S$ is reduced.

The standard deviation $\sigma$ is calculated at each grid point through all windstorms and over both seasons to ensure that the
two seasons can be compared to each other. By multiplying with the standard deviation, the sensitivity $S$ obtains the same units as the response function. Therefore, the sensitivity can be interpreted as how the response function (the minimum MSLP / maximum 10-m wind gust) changes when there is an increase of one standard deviation in the precursor field.

The sign of the sensitivity is determined only by the regression slope $m$ and the magnitude is effected by all three components in Eq. (1). Negative sensitivity means that the precursor $x$ and the response function $J$ are negatively correlated. Regarding
the response function of the minimum MSLP, this means that if the precursor increases it leads to a decrease in the minimum MSLP i.e. to a stronger windstorm in terms of MSLP. Likewise, regarding the response function of the maximum 10-m wind gust, negative sensitivity means that if the precursor increases it leads to a decrease in the maximum wind gust. However, this implies that the windstorm gets weaker in terms of wind gusts. This is important to note when comparing the sensitivities of these response functions that the sign of the sensitivity indicates contrary evolution to the windstorm.

In our study, we examine four precursors: 850-hPa potential temperature anomaly, TCWV, 300-hPa wind speed and 300-hPa PV. We selected these precursors because the 850-hPa potential temperature gradient is a main driver of extratropical cyclones through baroclinicity. TCWV allows us to investigate the availability of moisture that may influence diabatic processes which further may affect the intensity of an extratropical cyclone. The 300-hPa wind speed describes the jet stream and the location of an extratropical cyclone in relation to the jet stream is known to influence to the extratropical cyclone development. Lastly, the
300-hPa PV was chosen because the surface cyclone can intensify when it interacts with an upper-level trough which can be interpreted from the upper-level PV field. The 850-hPa potential temperature is used as an anomaly to assess the strength of the temperature gradient. The anomaly is calculated in each individual windstorm separately by first area averaging the 850-hPa potential temperature over the 18° radius composite and then subtracting this mean from each grid point in that windstorm. We investigate the precursor fields 24, 48 and 72 h before the time of the minimum MSLP or the maximum 10-m wind gust as
defined in Sect. 3.2. These offset times are relevant because they occur at the typical range of extratropical cyclone predictability and evolution. In addition, since the ensemble sensitivity method uses linear correlations the sensitivities are most likely valid at short lead times whereas at much longer lead times the non-linear affects would dominate.



## 4 Annual and monthly frequencies of extratropical cyclones, windstorms and non-windstorms

The annual frequency of extratropical cyclones during 1980–2019 shows a large year-to-year variability in all three cyclone classes (Fig. 2). There are on average 149 extratropical cyclones every year that reach Northern Europe (Fig. 2a) from which on average 35 cyclones i.e. 23 % are windstorms (Fig. 2c). Although the threshold for windstorms is the 90th percentile of wind gusts, the wind gust distribution can have multiple wind gust values per cyclone since we include all wind gust values at each time step when the cyclone centre is inside the Northern Europe box. Hence, the ratio of windstorms can be higher than 10 %. The 40-year linear trends are not statistically significant in any of the three cyclones classes by using Mann-Kendall test at 5 % level. However, there is a decreasing trend (-3.7 cyclones per decade) in all extratropical cyclones (Fig. 2a and 2b) with a p-value of 0.067 and therefore the trend would be significant with a 7 % level or higher. However, some time periods stand out with high/low annual number of extratropical cyclones. In the 1980s, there were overall more extratropical cyclones compared to the 40-year average (Fig. 2b) which was mainly due to the increased numbers of non-windstorms (Fig. 2f). In contrast, a longer term drop in all extratropical cyclones (Fig. 2b) occurred between 1995–2005, during which time the number of windstorms (Fig. 2d) decreased more than the number of non-windstorms. The highest annual numbers of windstorms occurred in 1983 and 1990 (Fig. 2c and 2d).

In addition to large inter-annual variability, we also investigated the seasonal cycle of the number of extratropical cyclones, windstorms and non-windstorms (Fig. 3). Regarding all extratropical cyclones in Northern Europe (Fig. 3a), the months from October to January and April have the highest median between 13–15 extratropical cyclones per month which means that on average almost every other day in these months an extratropical cyclone travels through Northern Europe. The variability of extratropical cyclone numbers within each month (length of whiskers) is the highest during autumn and winter. Nevertheless, the monthly variation in the number of all extratropical cyclones does not have a significantly large seasonal cycle. In contrast, the number of windstorms (Fig. 3b) shows a strong annual cycle with more windstorms in the cold season and less in the warm season. The highest median values of 5–6 windstorms per month are in January and December whereas in May, June and July, there are on average no windstorms at all. The variation of windstorm numbers within the cold season months is, however, quite large; for example in November, the number of windstorms ranges from zero to 14. The seasonal cycle for non-windstorms (Fig. 3c) is the opposite than for windstorms with the highest number of non-windstorms occurring during the warm season from April to August. Therefore, we can conclude that while windstorms are more common in winter and extratropical cyclones in summer are weaker, the overall number of extratropical cyclones per month in Northern Europe does not differ considerably depending on season.

## 5 Characteristics of extratropical cyclones, windstorms and non-windstorms

The genesis and track densities are computed by the TRACK code using the spherical kernel method (Hodges, 1996) and the unit of the densities is number of cyclones per month per unit area where the unit area is $5°$ radius spherical cap ($\approx 10^6$ km$^2$). Because of the $5°$ spherical cap in the unit, the cyclone numbers in the densities are larger than the absolute cyclone numbers in Sect. 4. In both seasons, the genesis region of all extratropical cyclones that affect Northern Europe extends from the east



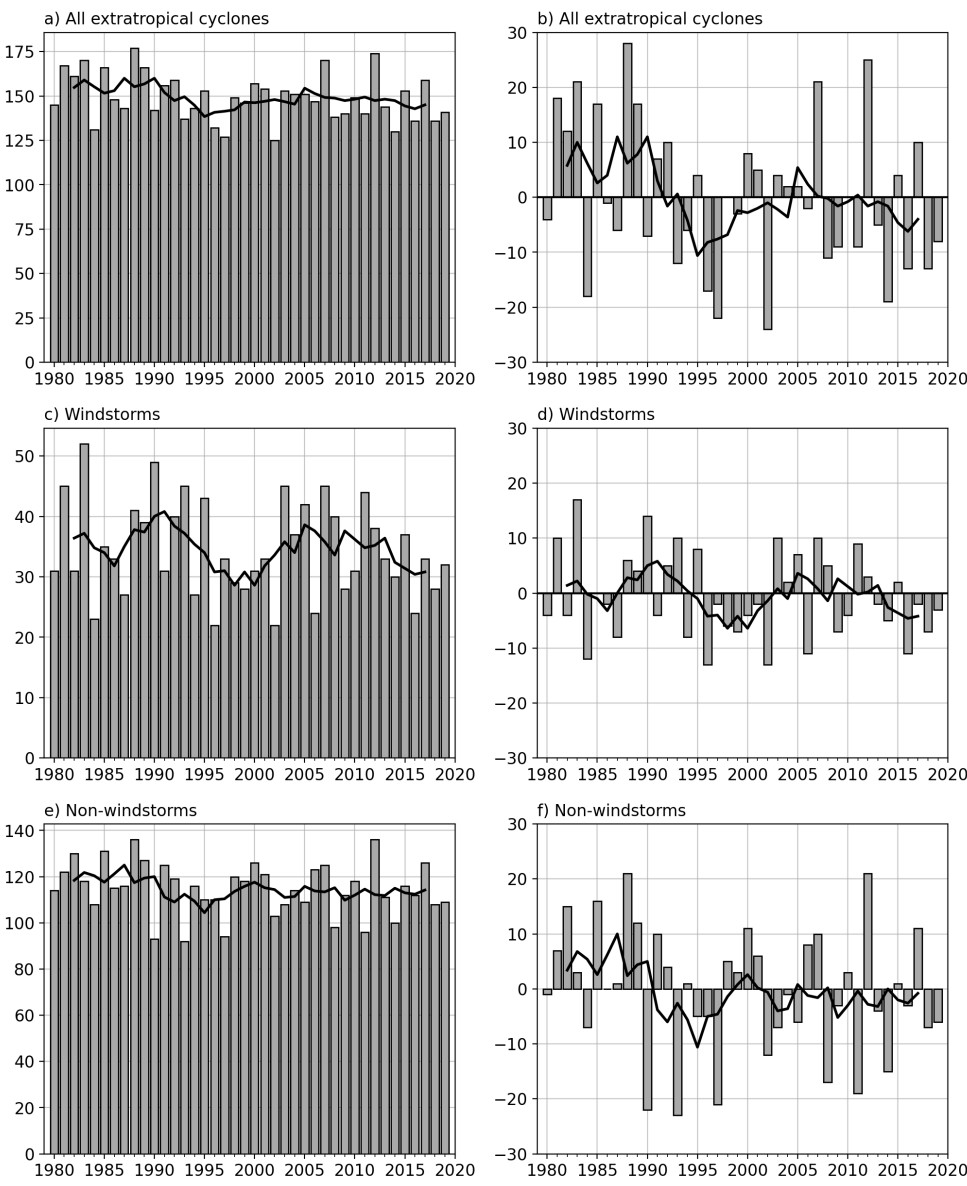

**Figure 2.** Annual numbers (left column) and absolute anomalies (annual value minus 40-year mean value; right column) of extratropical cyclones in Northern Europe in 1980–2019. Bars show the annual values and lines are 5-year running means. a,b) all extratropical cyclones, c,d) windstorms, and e,f) non-windstorms.

coast of the United States to western Russia and from the Greenland Sea to the Mediterranean Sea (Fig. 4a and 4b). Most of the extratropical cyclones in the warm season originate over the land areas inside the Northern Europe box (Fig. 4b). In the cold season (Fig. 4a), the most dense genesis region is broader and, in addition to the land areas in Northern Europe, most of





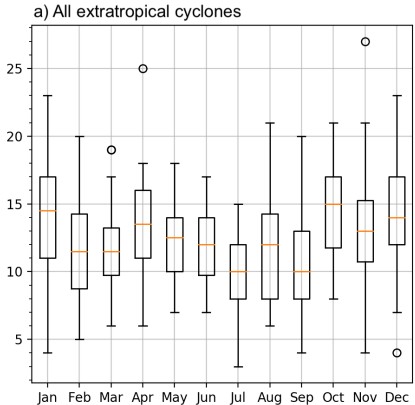 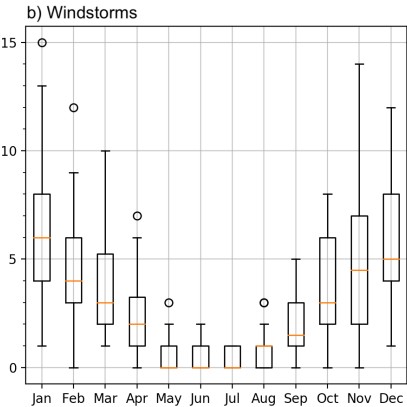 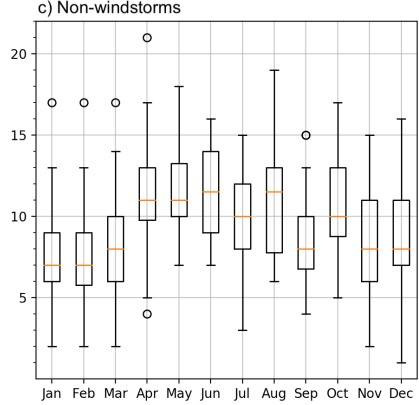

**Figure 3.** Monthly distribution of cyclones in Northern Europe. Orange lines are medians, boxes show first and third quartiles, and whiskers have first value greater than and last value less than Q1 - 1.5*IQR and Q3 + 1.5*IQR, respectively, where Q1 and Q3 are the first and third quartiles and IQR the interquartile range Q3 - Q1. a) all extratropical cyclones, b) windstorms, and c) non-windstorms.

the extratropical cyclones originate from the Norwegian Sea and the Barents Sea. The genesis over the east coast of the United

States is more poleward in the warm season than in the cold season which is in agreement with Priestley et al. (2020) who found that extratropical cyclones occur more equatorward in the winter than in the summer in the eastern North Atlantic.

When considering windstorms and non-windstorms separately, there are distinct differences. Most of the non-windstorms originate over land areas in Northern Europe (Fig. 4e and 4f) whereas windstorms have their genesis mostly over the Norwegian Sea and the Barents Sea (Fig. 4c and 4d). This also means that windstorms tend to originate further away from Northern Europe

than non-windstorms. In addition, non-windstorms can originate over Southern Europe while windstorms do not. In the cold season, the genesis region of windstorms is more zonally spread (Fig. 4c) whereas non-windstorms are more meridionally spread (Fig. 4e). In contrast, in the warm season the genesis region of non-windstorms is zonally spread and includes tracks from the eastern United States (Fig. 4f).

The track densities of all extratropical cyclones are, on a large scale, similar in both seasons, and the highest track density

occurs inside the Northern Europe box (Fig. 5a and 5b). Cold season extratropical cyclones have high track densities over the eastern part of Northern Europe and over the Barents Sea (Fig. 5a). There are somewhat less tracks over the Scandinavian Mountains which is likely caused by the Scandinavian Mountains that can, to some degree, modify the track of passing extratropical cyclones and even split the cyclone to two low centres (Grönås, 1997). When comparing windstorms and non-windstorms, it is evident that most windstorms have their cyclone centre at higher latitudes than non-windstorms (Fig. 5c-f).

The difference is more pronounced in the cold season where windstorms mostly occur over the Barents Sea (Fig. 5c) while non-windstorms over the southern parts of Northern Europe (Fig. 5e).

The majority of extratropical cyclones, regardless of the cyclone class (windstorm / non-windstorm) or season, propagate towards the east inside the Northern Europe box (not shown). However, each year there is a small portion of extratropical cyclones that propagate towards the west, and this is slightly more visible in the warm season than in the cold season. This

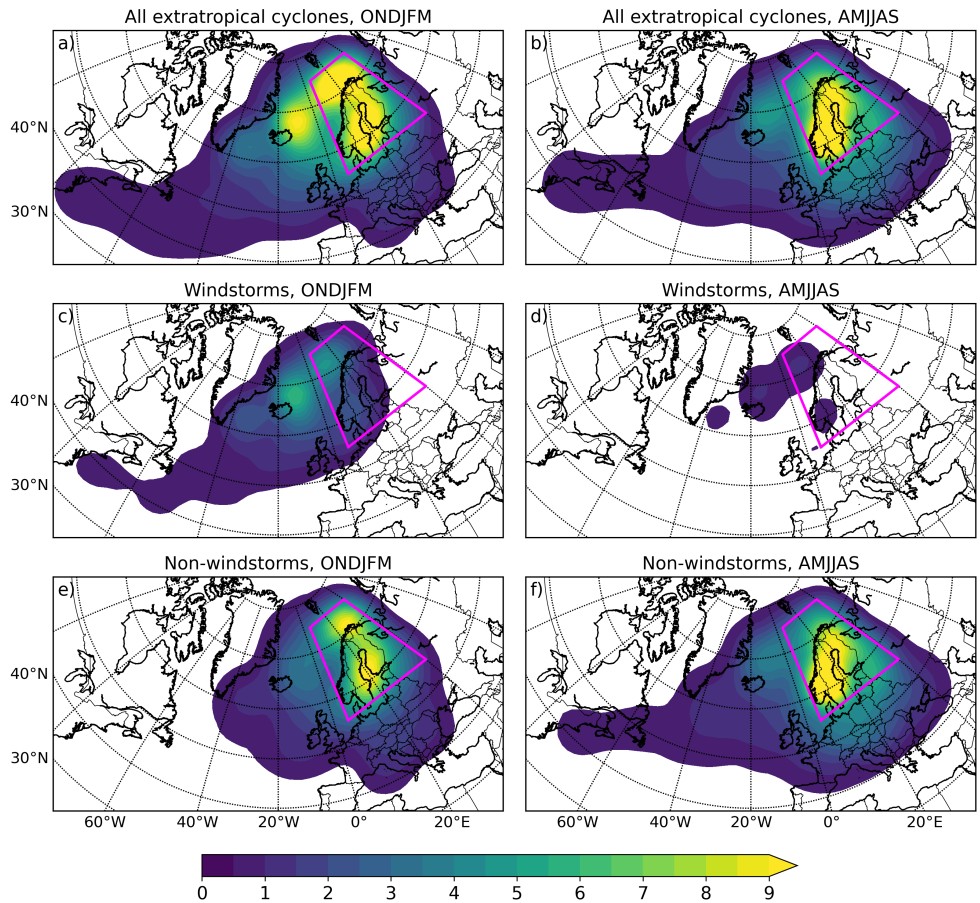

**Figure 4.** Genesis density (number of cyclones per month per $5°$ spherical cap $\approx 10^6$ km$^2$) of a,b) all extratropical cyclones, c,d) windstorms, and e,f) non-windstorms. The magenta box shows the Northern Europe box where the tracks in this study are filtered to occur.

is also seen in the genesis density (Fig. 4) where some extratropical cyclones originate on the eastern edge of the Northern Europe box.

     The median duration of the cyclone lifetime with all extratropical cyclones is 3.4 days in the cold season and 3.6 days in the warm season (not shown). Considering windstorms, the median lifetime is 4 days in both the cold and warm seasons while non-windstorms have a longer median lifetime in the warm season (3.6 days) than in the cold season (3 days, not shown). 265    Therefore, extratropical cyclones in Northern Europe are on average longer lived in the warm season compared to the cold season and in windstorms compared to non-windstorms. The lifetime of extratropical cyclones, especially in the extremely long lasting cases, can also be interpreted from the width of the histograms in Fig. 6. In the warm season the histogram is wider (from -13 to 17 days, Fig. 6b) indicating that the lifetimes are longer than in the cold season (from -9 to 10 days, Fig. 6a).

     Figure 6 shows the maximum 10-m wind gusts in Northern Europe (i.e. the maximum 10-m wind gust when the cyclone 270    centre is inside the Northern Europe box) in relation to the time of the minimum MSLP. The highest wind gust values on the



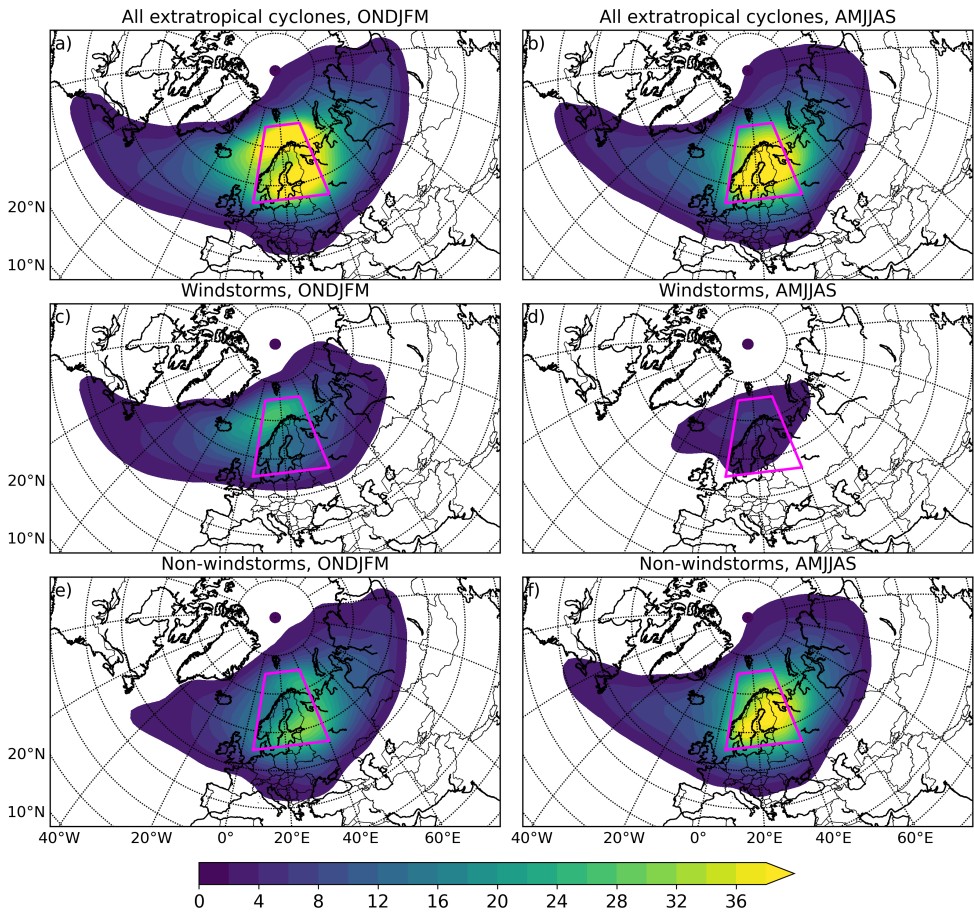

**Figure 5.** Track density (number of cyclones per month per $5°$ spherical cap $\approx 10^6$ km$^2$) of a,b) all extratropical cyclones, c,d) windstorms, and e,f) non-windstorms. The magenta box shows the Northern Europe box where the tracks in this study are filtered to occur.

y-axis are concentrated close to zero on the x-axis which means that the maximum wind gusts occur close to the time of the minimum MSLP. The median occurrence time of the maximum gusts relative to the time of the minimum MSLP is 0 h for all extratropical cyclones and non-windstorms and +3 h for windstorms. There is no differences between the seasons on the median occurrence times. Overall, the occurrence time of the wind gusts is practically very close to the minimum MSLP and

the difference between windstorms and non-windstorms is small. In the cold season, the maximum 10-m wind gusts are higher (up to 52 m s$^{-1}$) than in the warm season (up to 38 m s$^{-1}$), which was also seen in the wind gust distributions (Fig. 1). Figure 6 also shows that while the wind gusts are the strongest at the time of the minimum MSLP, after that the wind gusts decrease. This happens especially in windstorms (Fig. 6c and 6d) where a negative tilt is visible after the time of the minimum MSLP. The negative tilt is not evident in non-windstorms (Fig. 6e and 6f) but non-windstorms by definition do not have strong wind

gusts (there are no wind gust values above 27.2 m s$^{-1}$).



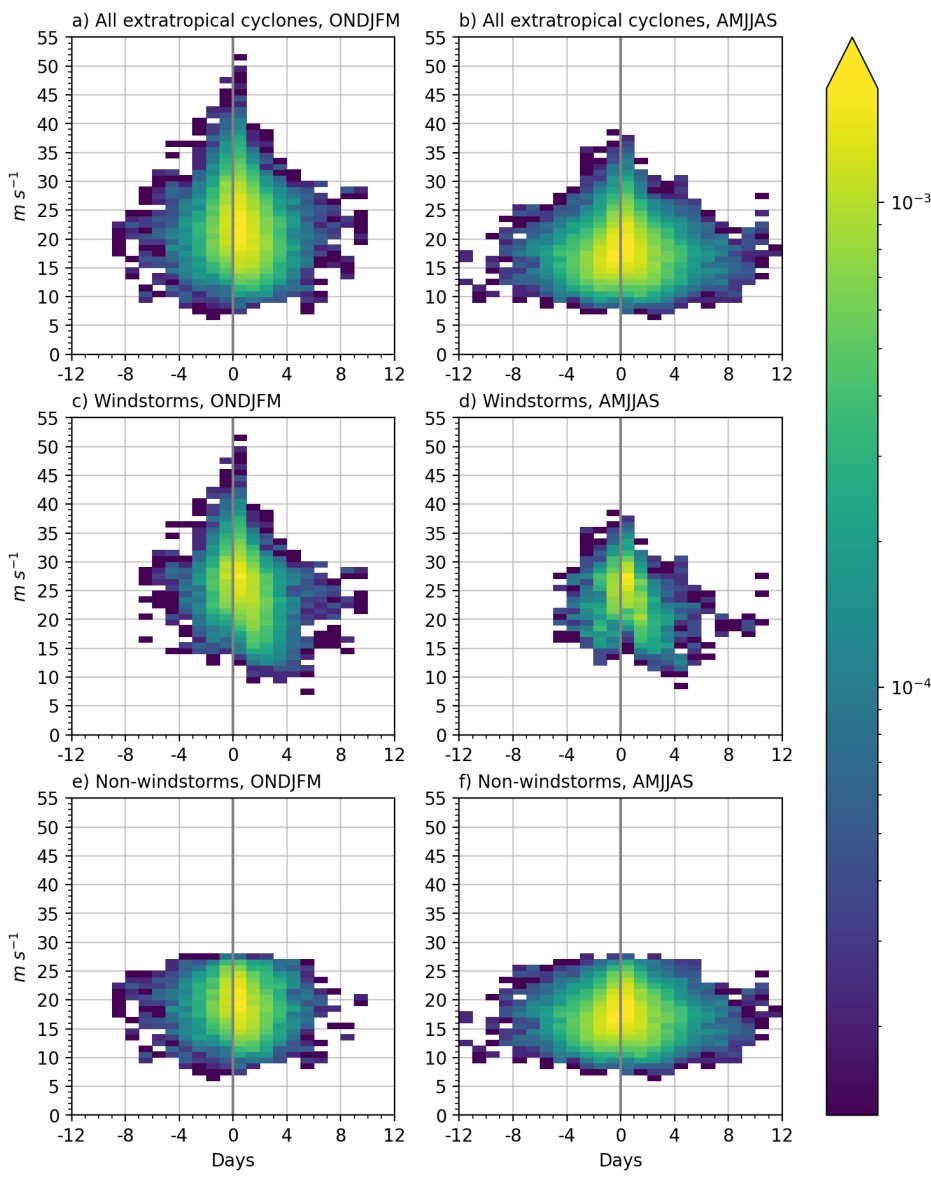

**Figure 6.** Normalized frequency of maximum 10-m wind gust (m s$^{-1}$) in Northern Europe (i.e. the maximum 10-m wind gust when the cyclone centre is inside the Northern Europe box) in relation to the time of the minimum mean sea level pressure (MSLP) during the cold season (left panel) and the warm season (right panel): a,b) all extratropical cyclones, c,d) windstorms, and e,f) non-windstorms. Zero on the x-axis indicates the time when the minimum MSLP during the track is obtained and negative/positive values indicate days before/after that time. The frequency is normalized by the total number of the points in the tracks, and the colors are in a logarithmic scale. There are three data points with x-axis values over 12 days which are cut from the figure.



## 6  Spatial and temporal evolution of maximum wind gusts, MSLP and frontal structure in windstorms

The composites of 10-m wind gust, MSLP and 850-hPa potential temperature are shown in Fig. 7 for the cold season and in Fig. 8 for the warm season and these fields allow us to determine the spatial structure of windstorms in Northern Europe. The offset times are relative to the time of the minimum MSLP of each windstorm (defined in Sect. 3.2). We investigate here the

offset times at -48 h, -24 h, 0 h and +24 h since Fig. 6 shows that these times cover the evolution before and after the strongest wind gusts and additionally include enough windstorms to ensure a reasonable amount of cyclones in each composite; at the 0 h offset time there are 1129 windstorms in the cold season and 245 windstorms in the warm season.

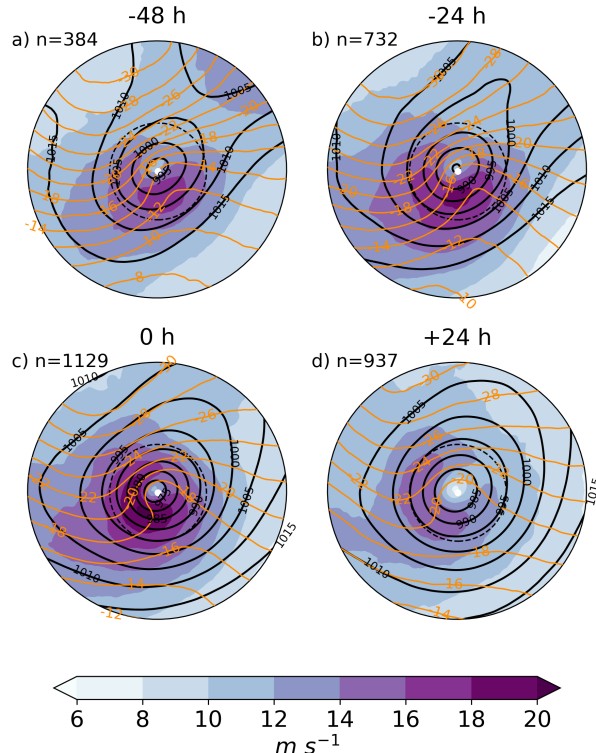

**Figure 7.** The maximum 10-m wind gust (colours, m s$^{-1}$), mean sea level pressure (black contours, 5 hPa interval) and 850-hPa potential temperature (orange contours, 2 °C interval) composites of windstorms during the cold season (October–March). The offset times are relative to the time of the minimum mean sea level pressure: a) -48 h, b) -24 h, c) 0 h, and d) +24 h. The number in each panel is the number of individual windstorms in each composite. The radius of the plots is 18° and the 6° radius is marked with a dashed circle.

Already 48 h before the minimum MSLP (Fig. 7a and 8a), a closed low pressure centre is evident in both seasons. The cold season composite has a deeper central pressure (994 hPa) and a stronger pressure gradient than the warm season composite

(998 hPa). The frontal structure is visible from the 850-hPa potential temperature composite with a warm front downstream of the cyclone centre and a cold front upstream of the cyclone centre. The strongest wind gusts occur in the warm sector in the



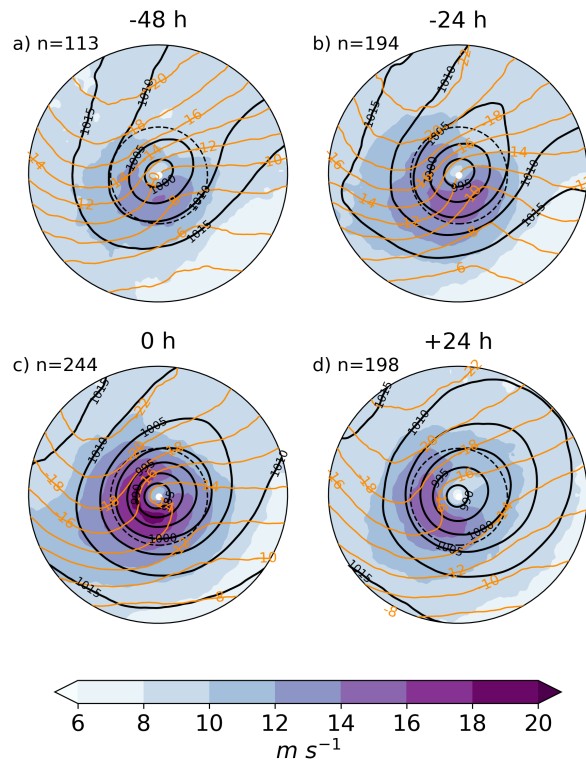

**Figure 8.** The maximum 10-m wind gust (colours, m s$^{-1}$), mean sea level pressure (black contours, 5 hPa interval) and 850-hPa potential temperature (orange contours, 2 °C interval) composites of windstorms during the warm season (April–September). The offset times are relative to the time of the minimum mean sea level pressure: a) -48 h, b) -24 h, c) 0 h, and d) +24 h. The number in each panel is the number of individual windstorms in each composite. The radius of the plots is 18° and the 6° radius is marked with a dashed circle.

region of the strongest pressure gradient. In addition, the strongest wind gusts are located within a 6° radius from the cyclone centre which supports the choice of selecting the maximum wind gust within 6 degrees. The strongest composite gusts are 17.2 m s$^{-1}$ in the cold season and 14.2 m s$^{-1}$ in the warm season. The region of the strongest gusts is more widely spread in the

cold season than in the warm season. This indicates that the windstorms in the cold season are larger in spatial scale than in the warm season.

At 24 h before the minimum MSLP (Fig. 7b and 8b), the central pressure decreases and the pressure and frontal gradients increase compared to the -48 h offset time. The wind gusts get stronger while the maximum gust location spreads extending from the warm sector to the cold front. At the time of the minimum MSLP (Fig. 7c and 8c), the central pressure in the cold

season is 972 hPa and in the warm season 981 hPa, and the pressure gradient is the strongest equatorward of the cyclone centre. The warm front starts to wrap around the cyclone centre indicating that the windstorm is reaching its mature stage. The wind gusts are the strongest at this offset time and the maximum gust values are 20.5 m s$^{-1}$ in the cold season and 18.6 m s$^{-1}$ in the warm season. The location of the maximum wind gusts occurs in a region extending from the warm sector to behind the





cold front where the pressure gradient is the strongest. The maximum wind gusts are still within a 6° radius from the cyclone

centre. At 24 h after the minimum MSLP (Fig. 7d and 8d), the central pressure has risen and the frontal structure weakened. The maximum wind gusts have decreased and now occur behind the cold front.

To conclude, the location of the maximum wind gusts moves from the warm sector to behind the cold front following the location of the strongest pressure gradient during the evolution of the composite windstorm. The wind gusts, pressure gradient, and potential temperature gradient are all stronger, and the minimum MSLP deeper, in the cold season than in the warm season.

In addition, the spatial scale of the windstorm is larger in the cold season than in the warm season.

## 7    Precursors to windstorms

In this section, we used the ensemble sensitivity analysis to investigate how two response functions, 1) the minimum MSLP and 2) the maximum 10-m wind gust, are sensitive to four different precursors at different offset times. The precursors we examine are 850-hPa potential temperature anomaly, TCWV, 300-hPa wind speed and 300-hPa PV, and these are analysed 72

h, 48 h and 24 h before the offset time.

### 7.1    Sensitivity of the minimum MSLP

The offset time in the minimum MSLP sensitivities is the time when the minimum MSLP occurs during the whole lifetime of the windstorm (same as in the composites in Section 6). The 850-hPa potential temperature anomaly composites (contours in Fig. 9) show similar frontal structures and seasonal differences as in Fig. 7 and 8 which show the absolute values. In the cold

season, the sensitivity to the 850-hPa potential temperature anomaly has a dipole pattern with negative sensitivity in the warm sector and positive sensitivity in the cold sector north of the cyclone centre. Negative sensitivity in the warm sector means that an increase in the 850-hPa potential temperature anomaly in this location leads to a lower minimum MSLP. The strongest negative sensitivities at -48 h (Fig. 9c) show that a one standard deviation increase in the 850-hPa potential temperature anomaly decreases the minimum MSLP by more than 13 hPa. Likewise, positive sensitivity in the cold sector means that an increase

in the 850-hPa potential temperature anomaly in this location leads to a higher minimum MSLP which, in other words, means that a decrease in temperature in cold sector leads to a lower minimum MSLP. The strongest positive sensitivities at -72h (Fig. 9a) and -48 h (Fig. 9c) show that a one standard deviation increase in the 850-hPa potential temperature anomaly increases the minimum MSLP by 10 hPa. This implies that a stronger temperature gradient (i.e. a warmer warm sector and a colder cold sector) leads to a deeper minimum MSLP and hence to a stronger windstorm in terms of MSLP. This is in agreement

with studies that have shown that stronger baroclinicity, and thus temperature gradient, leads to a stronger extratropical cyclone (Tierney et al., 2018; Rantanen et al., 2019). The dipole pattern is evident in the cold season at all offset times with the strongest sensitivities at -48 h. In the warm season, the dipole pattern is visible at -48 h (Fig. 9d) and -24 h (Fig. 9f) offset times, and the sensitivity at -48 h is distinctly higher than at -24 h. The minimum MSLP is not sensitive to the 850-hPa potential temperature anomaly at -72 h in the warm season (Fig. 9b).



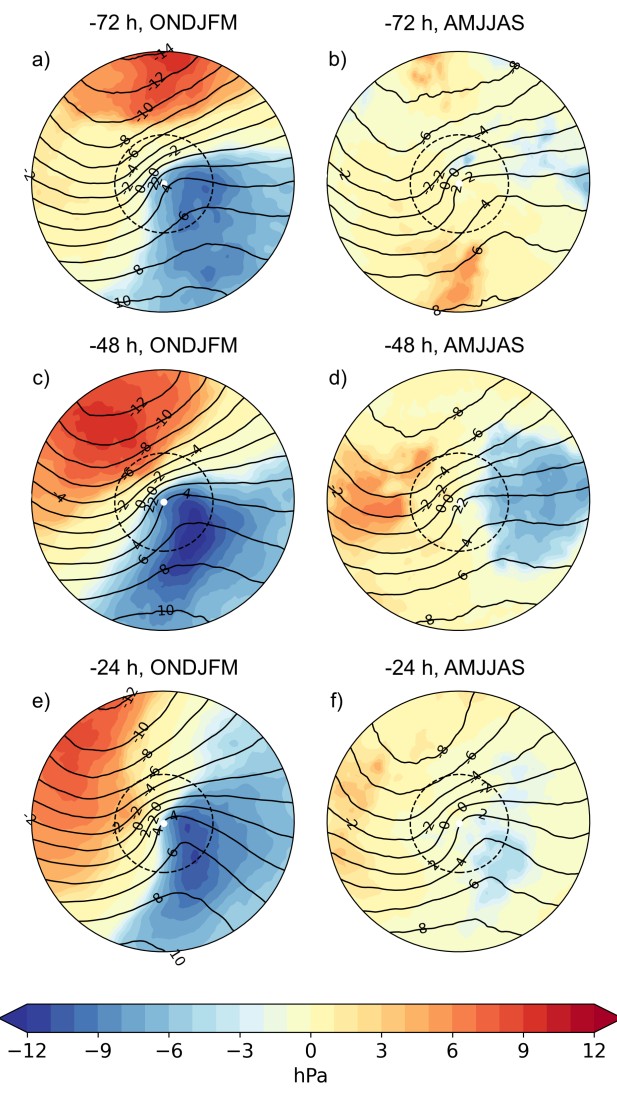

**Figure 9.** Sensitivity of the minimum MSLP to the 850-hPa potential temperature anomaly (colours, hPa), and the composite mean of 850-hPa potential temperature anomaly (contours, 2 °C interval) a,b) 72 h, c,d) 48 h, and e,f) 24 h before the occurrence of the minimum MSLP. Note that the color scale here is double than in the other sensitivity figures. The radius of the plots is 18° and the 6° radius is marked with a dashed circle.

The TCWV composites (contours in Fig. 10) have a similar pattern than the 850-hPa temperature anomaly composites and the highest amount of moisture is found south of the cyclone centre in the warm sector. The negative sensitivities south and downstream of the cyclone centre mean that more moisture in the warm sector leads to a deeper minimum MSLP. The sensitivities are higher in the cold season than in the warm season and have their largest values at -48 h offset time in both seasons. The highest sensitivity is in the cold season at -48 h offset time (Fig. 10c) where a one standard deviation increase in



TCWV decreases the minimum MSLP by 6 hPa. These results can be interpreted in a way that more moisture leads to more precipitation and this leads to more diabatic heating. Furthermore, diabatic heating can then produce low-level positive PV and this can further intensify the windstorm. Our result is consistent with Dacre et al. (2019) who similarly used ensemble sensitivity analysis to show that downstream moisture (TCWV) leads to an increase in total precipitation.

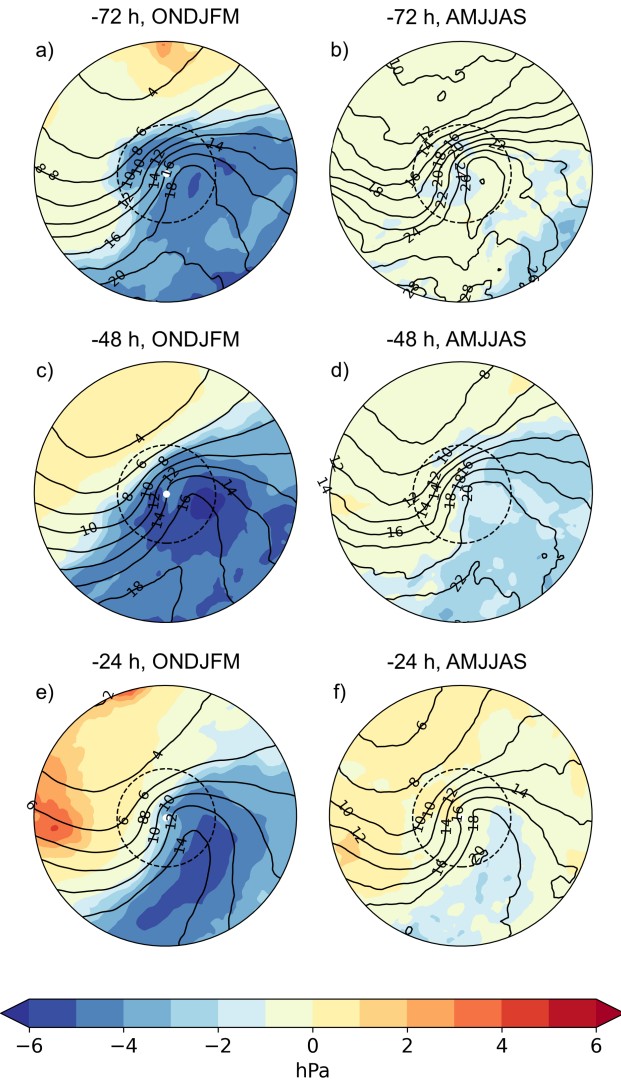

**Figure 10.** Sensitivity of the minimum MSLP to the TCWV (colours, hPa), and the composite mean of TCWV (contours, $2 \, \text{kg m}^{-2}$ interval) a,b) 72 h, c,d) 48 h, and e,f) 24 h before the occurrence of the minimum MSLP. The radius of the plots is $18°$ and the $6°$ radius is marked with a dashed circle.

The 300-hPa wind speed composites (contours in Fig. 11) show that the cyclone centre at -72 h is located in the centre of
the poleward side of the jet stream but at -48 h and -24 h the cyclone centre is located at the left exit of the jet stream. The





windstorm only starts strongly deepening at -48 h (not shown) when it moves into the left exit region (favourable region for cyclone development) and then 48 h later at 0 h, the windstorm has its maximum intensity in terms of MSLP. In the cold season at -72 h (Fig. 11a) and -48 h (Fig. 11c) offset times, the largest sensitivities are northeast of the cyclone centre. The strongest negative sensitivities in Fig. 11 mean that when the 300-hPa wind speeds increase by a one standard deviation it results in

a decrease in minimum MSLP by 6 hPa. This implies that the minimum MSLP of a windstorm is deeper when the 300-hPa jet stream is extended northeast of the cyclone centre. A situation when the 300-hPa wind speed is stronger northeast of the cyclone centre suggests that the jet stream is tilted and thus more meridional. Furthermore, this implies that although at -72 h the cyclone centre is not located at the left exit of the jet, it possibly is in a region of cyclonic curvature vorticity advection. The 300-hPa wind speed at -24 h (Fig. 11e) has large sensitivities over the whole jet stream meaning that the minimum MSLP

decreases when the jet stream is stronger. In the warm season, there is more variability between the offset times than was observed in the cold season. The -24 h (Fig. 11f) sensitivity has a similar pattern than in the cold season that a stronger jet stream leads to a deeper windstorm. While the precursor at -48 h (Fig. 11d) shows some sensitivity for the northeast extended jet stream, at -72 h (Fig. 11b) the field is more noisy and has no sensitivity. It is good to note that climatologically the jet stream is at a higher altitude in summer than in winter. However, the 300-hPa wind speed composites in the warm season do capture

the jet streak and the absolute 300-hPa wind speeds are not much weaker than in the cold season when the jet wind speeds are stronger.

The 300-hPa PV composites (contours in Fig. 12) show high PV values (> 2 PVU, indicating stratospheric air) that occur upstream and north of the cyclone centre denoting an upper-level trough. This westward tilt between a surface cyclone and an upper-level trough is usual in intensifying extratropical cyclones. An upper-level ridge (low PV values) is also seen downstream

of the cyclone centre with a PV gradient between the trough and the ridge which corresponds to a tilted tropopause and thus the location of the jet stream. In the cold season, all offset times show that there is positive sensitivity in the warm sector meaning that higher 300-hPa PV in that region leads to an increase in the minimum MSLP. Quantitatively, a one standard deviation increase in the 300-hPa PV leads to an increase in the minimum MSLP by up to 5 hPa. In other words, lower PV in the warm sector leads to a deeper windstorm. Lower PV in the warm sector implies a stronger upper-level ridge or higher tropopause

and this increases the PV gradient i.e. steepens the tropopause. This is in agreement with the 300-hPa wind speed (Fig. 11): a steeper tropopause is consistent with a stronger jet stream. In addition to being sensitive to a stronger upper-level ridge, at -24 h (Fig. 12e) the minimum MSLP is also sensitive to a stronger upper-level trough which similarly increases the PV gradient. In the warm season, the 300-hPa PV sensitivities are rather noisy and weak. This may imply that warm season windstorms are less controlled by the upper-level vorticity advection and more by the low-level thermal advection or diabatic heating (more

discussion on this in Sect. 8). Some sensitivity is visible to a stronger upper-level ridge at -48 h offset (Fig. 12d) and a stronger upper-level trough at -24 h offset (Fig. 12f) but mostly the minimum MSLP is not sensitive to the upper-level PV in warm season windstorms.



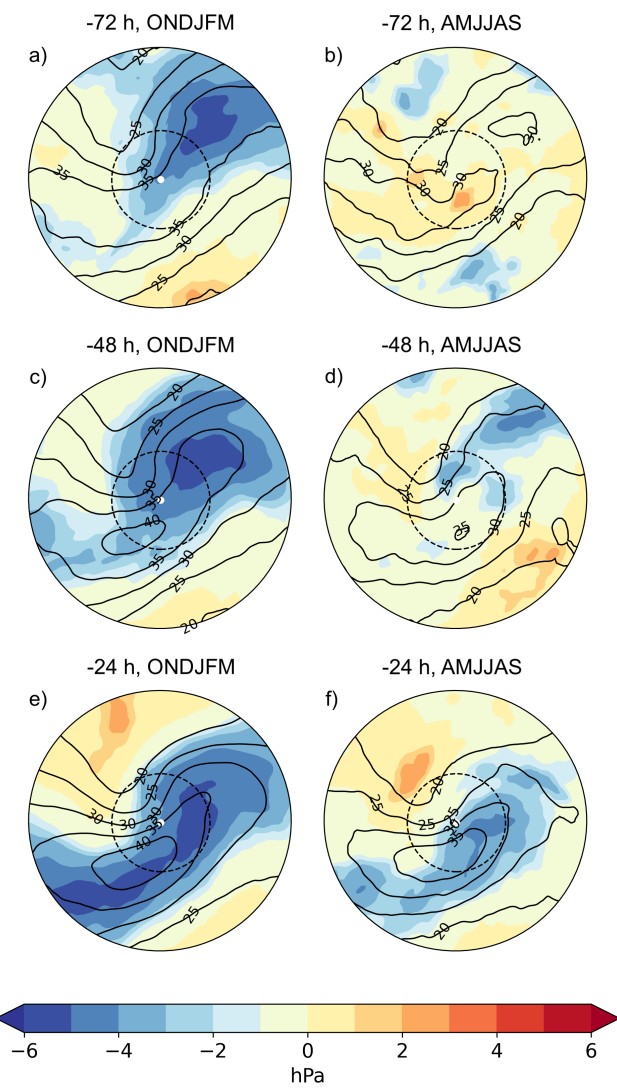

**Figure 11.** Sensitivity of the minimum MSLP to the 300-hPa wind speed (colours, hPa), and the composite mean of 300-hPa wind speed (contours, 5 m s$^{-1}$ interval) a,b) 72 h, c,d) 48 h, and e,f) 24 h before the occurrence of the minimum MSLP. The radius of the plots is 18° and the 6° radius is marked with a dashed circle.

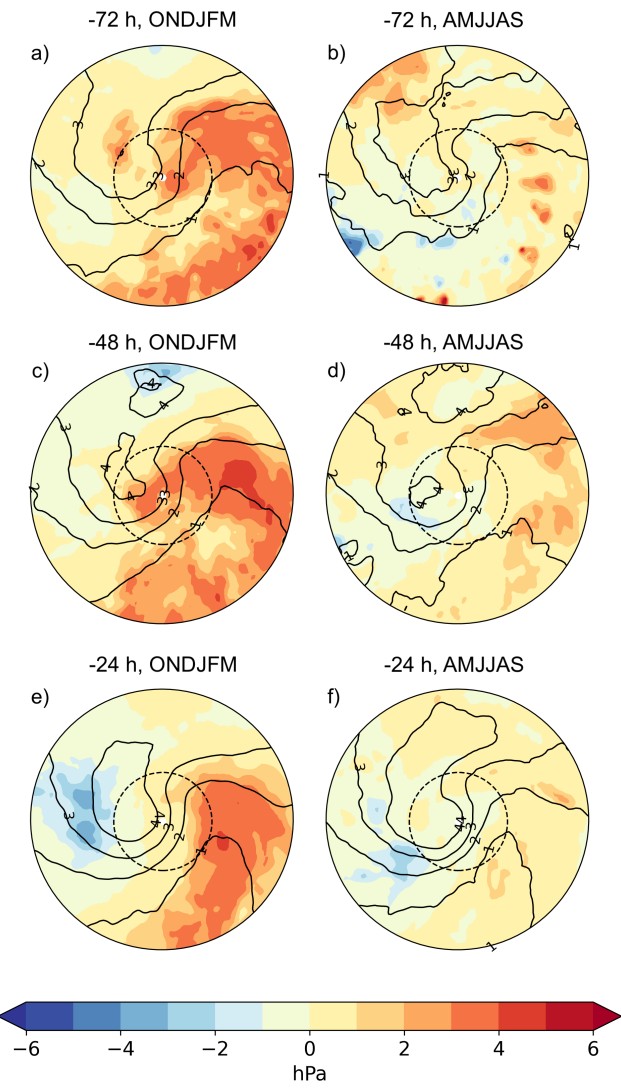

**Figure 12.** Sensitivity of the minimum MSLP to the 300-hPa PV (colours, hPa), and the composite mean of 300-hPa PV (contours, 1 PVU interval, 1 PVU = $1.0 \times 10^{-6}$ m$^2$ s$^{-1}$ K kg$^{-1}$) a,b) 72 h, c,d) 48 h, and e,f) 24 h before the occurrence of the minimum MSLP. The radius of the plots is $18°$ and the $6°$ radius is marked with a dashed circle.





## 7.2 Sensitivity of the maximum 10-m wind gust

In the maximum 10-m wind gust sensitivities, the offset time is the time when the maximum 10-m wind gust occurs when
the cyclone centre is inside the Northern Europe box. This means that the composite means are slightly different in this Sect.
compared to the previous Sect. 7.1. The 850-hPa potential temperature anomaly composites (contours in Fig. 13) reveal that the
frontal structure evolves within the 72 h before the maximum 10-m wind gust from a broad, large-scale temperature gradient
with the beginnings of a frontal wave (Figs. 13a and 13b) to clearer frontal wave patterns (Figs. 13e and 13f). The sensitivity
of the maximum 10-m wind gust has a similar dipole pattern than in the sensitivity of the minimum MSLP denoting that a
stronger temperature gradient leads to stronger gusts. This dipole pattern is visible in both seasons and all offset times. The
strongest sensitivities are in the cold season and at -72 h and -48 h offset times in two regions, in the warm sector and in the
cold sector. A one standard deviation increase in the 850-hPa potential temperature anomaly in the warm sector increases the
maximum wind gust by 2.4 m s$^{-1}$. A one standard deviation increase in the 850-hPa potential temperature in the cold sector
anomaly decreases the maximum wind gust by 2.0 m s$^{-1}$.

The TCWV composites (Fig. 14) are consistent with the frontal structure with the highest TCWV values found in the warm
sector. TCWV sensitivities show that in the cold season, more moisture south of the cyclone centre 72 h (Fig. 14a) and 48 h
(Fig. 14c) before the maximum 10-m wind gust occurrence leads to stronger gusts. As discussed above, moisture may lead
to diabatically produced low-level PV that deepens the minimum MSLP of the windstorm and hence increases the pressure
gradient. Stronger pressure gradient then leads to stronger wind gusts. The high sensitivities at -72 h are found in a broader
area than at -48 h where they are more concentrated near the cold front. The strongest sensitivities at -48 h offset (Fig. 14c)
indicate that a one standard deviation increase in TCWV in the warm sector increases the maximum wind gust by 1.6 m s$^{-1}$.
The TCWV sensitivities in the cold season at -24 h offset time and in the warm season at all offset times are weak, although
the sensitivity patterns are similar.

The 300-hPa wind speed composites (contours in Fig. 15) show that in the cold season, the cyclone centre is located at the
left exit region of the jet stream. The sensitivities in the cold season show that when the jet stream is further north (and more
downstream especially at -48 h) the wind gusts are stronger. A one standard deviation increase in the 300-hPa wind speed
north of the cyclone centre leads to an increase in the maximum wind gusts by up to 1.6 m s$^{-1}$. This suggests that stronger
wind gusts occur in cold season windstorms possibly when the jet stream is more poleward relative to the cyclone centre. The
maximum wind gust is more sensitive to the jet stream at -72 h (Fig. 15a) and -48 h (Fig. 15c) offset times than -24 h (Fig.
15e) offset time. In the warm season, the cyclone centre is located at the centre of the poleward side of the jet stream. The
warm season sensitivities have similar pattern than the cold season but they are weaker: a one standard deviation increase in
the 300-hPa wind speed in the warm season leads to an increase in the maximum wind gusts by up to 1.2 m s$^{-1}$.

Overall, the 300-hPa PV gives weak sensitivities to the maximum 10-m wind gust. In the cold season, -72 h (Fig. 16a) and
-48 h (Fig. 16c) offset times show that a stronger upper-level ridge (i.e. lower PV values downstream of the cyclone centre)
leads to slightly stronger gusts. A one standard deviation increase in the 300-hPa PV leads to a decrease in the maximum wind
gusts by 1.2 m s$^{-1}$. This is not visible at -24 h or at any offset times in the warm season. However, in the warm season at



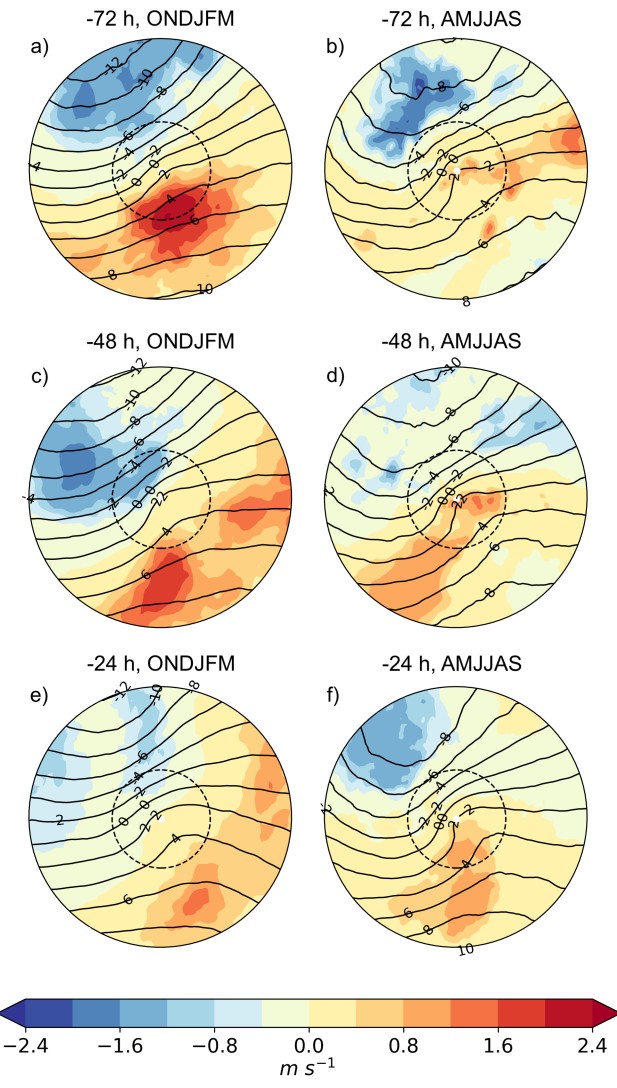

**Figure 13.** Sensitivity of the maximum 10-m wind gust to the 850-hPa potential temperature anomaly (colours, m s$^{-1}$), and the composite mean of 850-hPa potential temperature anomaly (contours, 2 °C interval) a,b) 72 h, c,d) 48 h, and e,f) 24 h before the occurrence of maximum 10-m wind gust in Northern Europe. The radius of the plots is 18° and the 6° radius is marked with a dashed circle.

-48 h offset time (Fig. 16d), there is weak sensitivity north and upstream of the cyclone centre where a one standard deviation increase in the 300-hPa PV (i.e. a stronger upper-level trough) leads to an increase of the maximum wind gusts by 0.8 m s$^{-1}$.



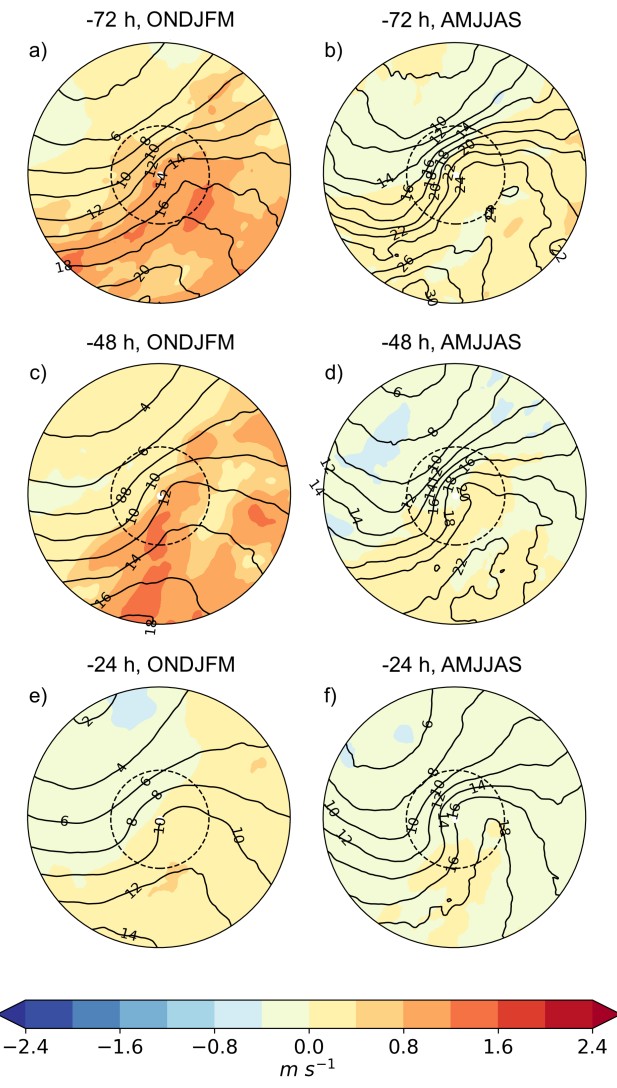

**Figure 14.** Sensitivity of the maximum 10-m wind gust to the TCWV (colours, m s$^{-1}$), and the composite mean of TCWV (contours, 2 kg m$^{-2}$ interval) a,b) 72 h, c,d) 48 h, and e,f) 24 h before the occurrence of maximum 10-m wind gust in Northern Europe. The radius of the plots is 18° and the 6° radius is marked with a dashed circle.



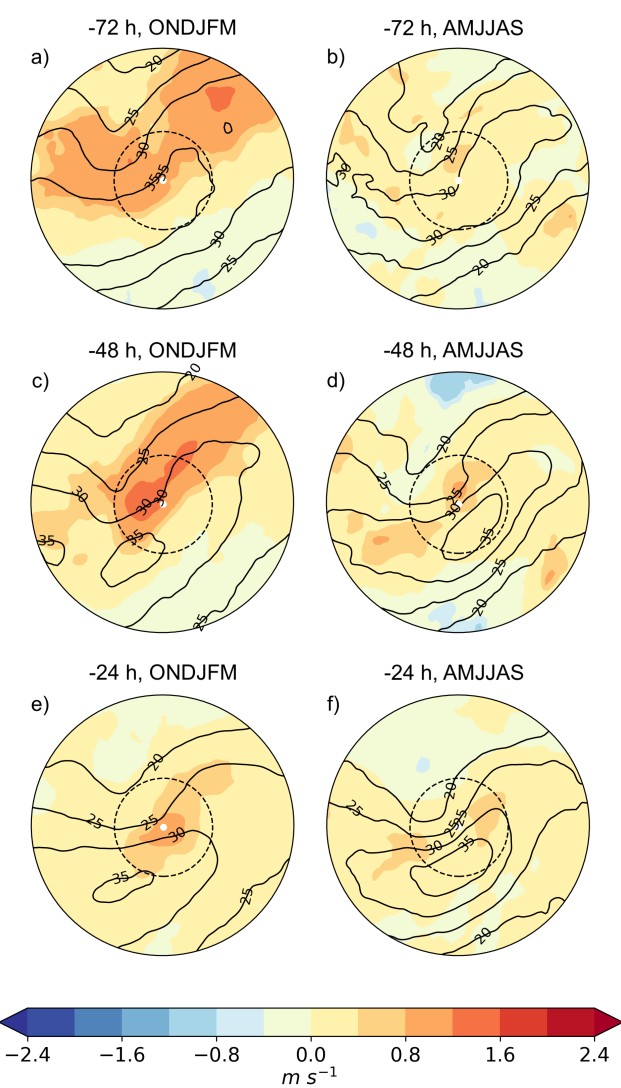

**Figure 15.** Sensitivity of the maximum 10-m wind gust to the 300-hPa wind speed (colours, m s$^{-1}$), and the composite mean of 300-hPa wind speed (contours, 5 m s$^{-1}$ interval) a,b) 72 h, c,d) 48 h, and e,f) 24 h before the occurrence of maximum 10-m wind gust in Northern Europe. The radius of the plots is 18° and the 6° radius is marked with a dashed circle.



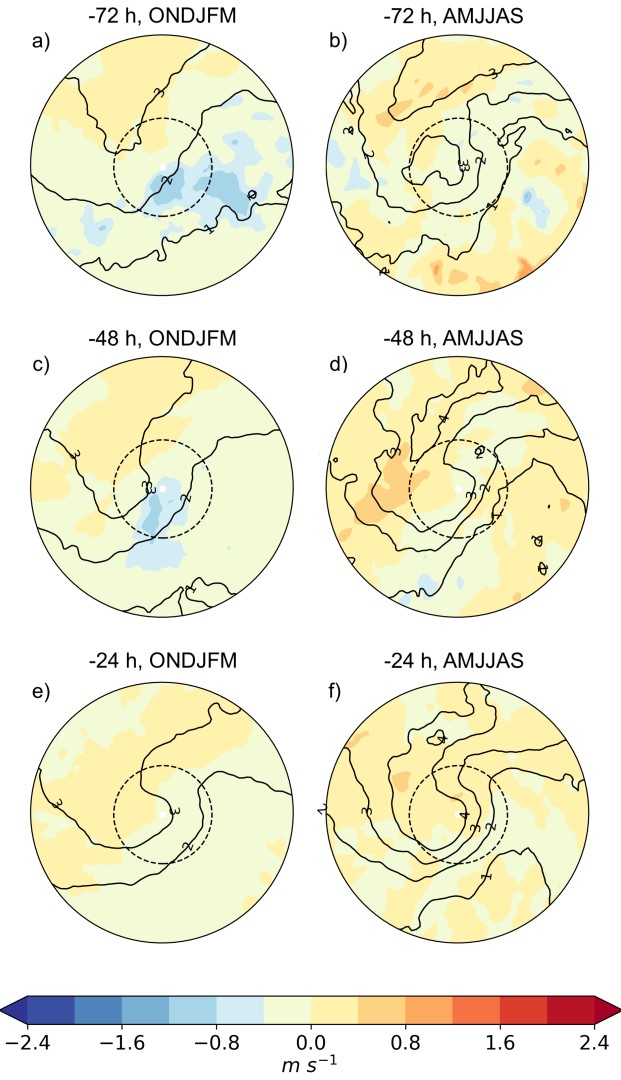

**Figure 16.** Sensitivity of the maximum 10-m wind gust to the 300-hPa PV (colours, m s$^{-1}$), and the composite mean of 300-hPa PV (contours, 1 PVU interval, 1 PVU = $1.0 \times 10^{-6}$ m$^2$ s$^{-1}$ K kg$^{-1}$) a,b) 72 h, c,d) 48 h, and e,f) 24 h before the occurrence of maximum 10-m wind gust in Northern Europe. The radius of the plots is 18° and the 6° radius is marked with a dashed circle.



## 8  Discussion and conclusions

This study investigated extratropical cyclones and windstorms in Northern Europe and their characteristics, spatial and temporal evolution, and precursors. We tracked all extratropical cyclones in Northern Europe during 1980–2019 and classified them to three classes: 1) all extratropical cyclones in Northern Europe, 2) windstorms (i.e. extratropical cyclones which have strong wind gusts in Northern Europe), and 3) non-windstorms (i.e. extratropical cyclones with no strong wind gusts in Northern Europe). We investigated the cold season (October–March) and the warm season (April–September) separately to identify

seasonal differences. We created cyclone composites to examine the cyclone evolution and used an ensemble sensitivity method to analyse precursors.

During the 40-year period, there is a large year-to-year variation in the annual number of extratropical cyclones, windstorms and non-windstorms in Northern Europe. The 40-year linear trends are not significant at 95 % level, however, there is a decreasing trend in all extratropical cyclones (-3.7 cyclones per decade) that is significant at 93 % level. Feser et al. (2015)

reviewed the long-term trends of extratropical cyclone numbers over the North Atlantic and northwestern Europe. They found that in their "Baltic Sea" region (covering the Baltic Sea, the Baltic states, and the major parts of Finland, Sweden and Norway) there is no consensus on the long-term trends in reanalysis, model or observation based studies. Nonetheless, the sign and magnitude of linear trends depend strongly on the chosen time period and region, and therefore comparing different studies and results is difficult. In addition, it is important to note that a low number of windstorms in a specific year or decade does not

mean that there could not be individual, strong windstorms. For example, our study shows that there were less windstorms than average between 1995–2005 while during this period a few extremely strong and damaging windstorms occurred in Northern Europe: storm Anatol in December 1999 (Ulbrich et al., 2001) and storm Gudrun in January 2005 (Suursaar et al., 2006).

We found the well known seasonality that windstorms are more common in winter and extratropical cyclones in the warm season are weaker. However, a more unexpected result was that the overall number of extratropical cyclones per month in

Northern Europe does not differ much between months. This is contradictory to the general claim that there are less extratropical cyclones in the Northern Hemisphere during summer than winter (see for example the review by Ulbrich et al. (2009)). However, this seasonal difference is most pronounced in the core of the North Atlantic storm track while at the end of the storm track, i.e. in Northern Europe, the difference is smaller (Hoskins and Hodges, 2019; Priestley et al., 2020). If considering the track densities in Northern Europe, Hoskins and Hodges (2019) show mainly no difference between winter and summer

and Priestley et al. (2020) show only a small difference. Therefore, our finding does agree with other extratropical cyclone climatologies but the other studies simply highlight more the general seasonality in the core of the storm track although the seasonality has regional differences.

We investigated a set of cyclone characteristics to detect possible differences between windstorms and non-windstorms and between the cold and warm seasons. Windstorms tend to originate and occur over the sea areas (in the Norwegian and Barents

Seas) while non-windstorms originate and occur mostly over land in Northern Europe. Partially, this is likely related to surface friction since we define windstorms based on their wind gusts and winds are stronger over smooth surfaces i.e. over sea areas. Moreover, the highest genesis densities of windstorms are co-located with the climatological left exit of the jet stream (Laurila



et al., 2021) which is a favourable location for cyclone development. Our result also means that windstorms occur at higher latitudes than non-windstorms. This agrees with our composites which showed that windstorms are located on the poleward side of the jet stream in both seasons. We additionally found that the maximum wind gusts occur approximately at the same time as the minimum MSLP regardless of the cyclone class or season.

The maximum wind gusts in windstorms move from the warm sector to behind the cold front during the cyclone evolution following the strongest pressure gradient. This shift in the maximum gust location during the cyclone evolution is in agreement with the shift in the maximum 900-hPa wind speeds of the 200 strongest extratropical cyclones in an aqua-planet study (Sinclair et al., 2020). In addition, the spatial patterns of MSLP and 850-hPa potential temperature in our composites are similar that are found in the composites of the 100 strongest extratropical cyclones in the Northern Hemisphere (Catto et al., 2010). This suggest that Northern Europe windstorms have similar spatial and temporal structures that are found elsewhere in the Northern Hemisphere. Our results show that the wind gusts, pressure gradient and temperature gradient are stronger and the minimum MSLP is deeper in the cold season than in the warm season. The cold season windstorms have their associated wind gusts covering a larger area and thus are spatially larger than warm season windstorms. This is relevant when informing and preparing for likely impacts yet it is important to note that our study only includes large-scale extratropical cyclones whereas other weather systems (e.g. thunderstorms) can also cause strong winds and impacts.

Lastly, we investigated precursors to Northern Europe windstorms for the minimum MSLP and the maximum 10-m wind gust. The first main conclusion of our precursor results is that cold season windstorms have higher sensitivities and thus are potentially easier to forecast than warm season windstorms in terms of both the minimum MSLP and the maximum wind gusts. Possible reasons for this are a higher case-to-case variability and more rare occurrence of warm season windstorms compared to cold season windstorms. The second main conclusion is that the minimum MSLP of a windstorm has higher sensitivities than its associated maximum wind gust. This is possibly caused by the different nature of these variables: MSLP is a larger scale field while wind gusts are more localised and turbulent. The third and last main conclusion is that the best precursor for Northern Europe windstorms is the 850-hPa potential temperature anomaly i.e. the temperature gradient. This precursor outperformed the TCWV (moisture), the 300-hPa wind speed (jet stream), and the 300-hPa PV (upper-level trough and tropopause steepness). This means that the 850-hPa temperature gradient has the strongest impact on the resulting minimum MSLP and the maximum wind gusts and thus the temperature gradient is an important variable to follow when forecasting windstorms in Northern Europe. If we apply our sensitivity results to climate change, the low-level temperature gradient is expected to decrease (IPCC, 2013) which based on our results would indicate weaker windstorms in terms of the minimum MSLP and the maximum wind gusts. However, this is counteracted by the atmospheric moisture which is predicted to increase (IPCC, 2013) and hence would, based on our results from the TCWV sensitivities, indicate stronger windstorms.

The 300-hPa PV shows sensitivities to the downstream ridge which is not found in Dacre and Gray (2013) who used the ensemble sensitivity analysis to investigate precursors to extratropical cyclones in the west and east North Atlantic. This difference between our and their results may be because of the difference in the geographical location: blocking is more common over Northern Europe / Russia than over the North Atlantic (Pelly and Hoskins, 2003; Rimbu and Lohmann, 2011). The 300-hPa PV also gives weak or no-existent sensitivities in the warm season. Petterssen and Smebye (1971) and Deveson



et al. (2002) developed a cyclone development classification based on the upper and lower level forcing mechanisms and they identified three types of cyclones. Type A cyclones develop due to lower level thermal advection in baroclinic regions without a pre-existing upper-level trough, type B cyclones develop due to upper-level vorticity advection where a pre-existing upper-level trough moves over an area of warm advection, and type C cyclones are characterized by strong latent heat release with initiation of strong upper-level forcing and weak low-level baroclinicity (Petterssen and Smebye, 1971; Deveson et al., 2002; Plant et al., 2003). Therefore, our result may indicate that warm season windstorms in Northern Europe are type A or type C cyclones since they are not dominated by upper-level forcing. Our finding that the TCWV gives weak sensitivities in the warm season was surprising since typically there is more diabatic heating in the warm season than in the cold season. However, this may be a result of a low number of windstorms and a high case-to-case variability in the warm season. Our results also showed that the sensitivities are generally higher at -48 h than at -24 h. One possible explanation could be that 24 h is too short time for the atmospheric available potential energy to be converted to kinetic energy, an energy conversion know as the Lorenz energy cycle (Lorenz, 1955).

Our study has certain limitations. There are multiple ways to classify extratropical cyclones and to define windstorms (Catto, 2016) and therefore, our results may depend on our windstorm definition. Due to different extratropical cyclone and windstorm definitions, direct comparison of our climatology to previous studies is not simple. We also made a subjective choice about what geographic area to study and our results are likely to be sensitive to our choice of the box over Northern Europe. The area we considered included sea areas and most windstorms we identified occurred over sea making our results less applicable to windstorms over land areas. Furthermore, our analysis was based solely on ERA5 reanalysis which may not accurately capture wind gusts due to the coarse resolution of the reanalysis (∼31 km) and also because wind gusts are parameterized, not directly resolved. Another limitation is that composites (and climatological studies overall) smooth the case-to-case variability and therefore there is still need for case studies to investigate different evolution paths in more detail. For example, storm Aila occurred in Finland in September 2020 and developed in the right entrance of the jet stream (Rantanen et al., 2021) which is contradictory to our result that Northern Europe windstorms occur poleward of the jet stream. On the other hand, storm Aila formed in an area of a strong temperature gradient (Rantanen et al., 2021) which agrees with our results.

Despite the limitations noted above, our composite and sensitivity results highlight the common features of windstorms in Northern Europe. This enhances our dynamical understanding of these potentially damaging weather systems and furthermore, provides valuable guidance, in the form of data based "conceptual models" to weather forecasters in the region.



# Appendix A

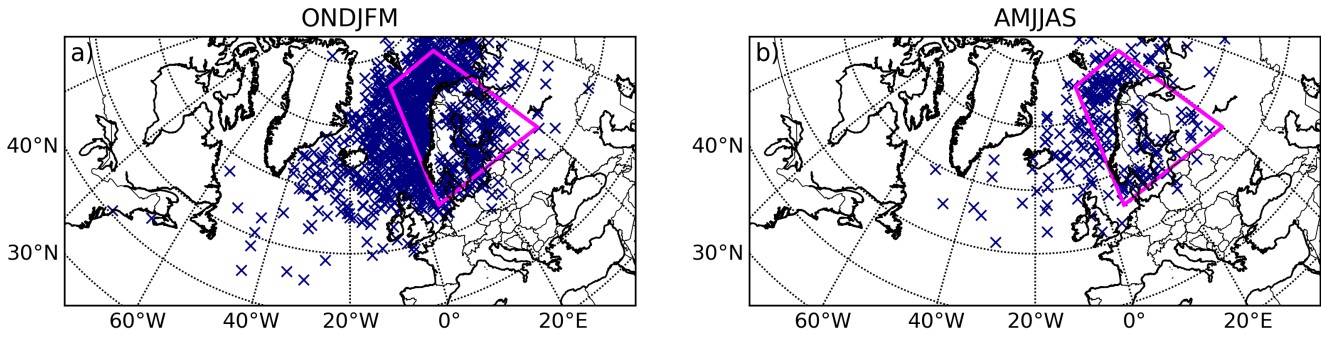

**Figure A1.** The locations of the minimum MSLP of windstorm tracks. The time of the minimum MSLP is the time when the absolute minimum of MSLP occurs along the cyclone track.

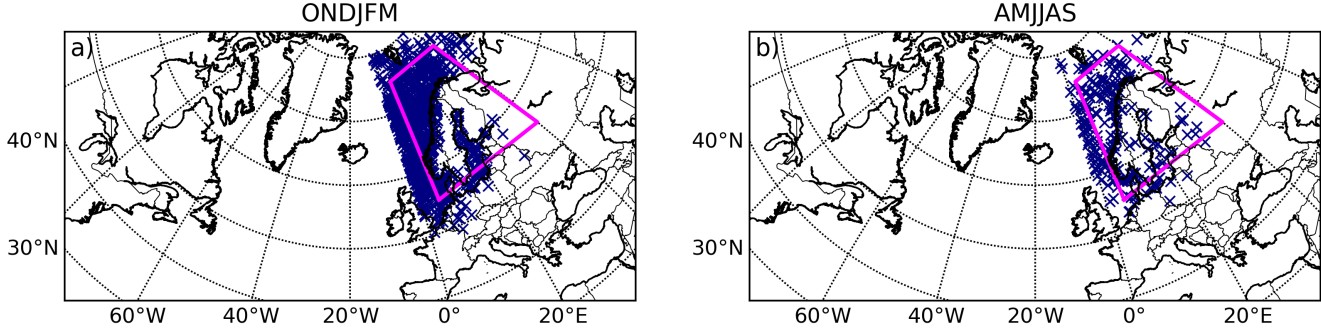

**Figure A2.** The locations of the maximum 10-m wind gust of windstorm tracks. The time of the maximum 10-m wind gust is the time when the maximum wind gust occurs while the cyclone centre is within the Northern Europe box.

*Code availability.* Information on how to obtain the cyclone identification and tracking algorithm (TRACK) can be found from http://www.nerc-essc.ac.uk/~kih/TRACK/Track.html.

*Author contributions.* TKL performed most of the data analysis, interpreted the results, drew the main conclusions, and wrote the majority of the paper. JC contributed to the data analysis of the wind speed distributions and the cyclone characteristics presented in sections 4, 5 and 6. VAS performed the cyclone tracking. VAS and HG conceived the study, provided guidance on the interpretation of the results, and led the overall scientific investigation. All authors provided comments on drafts of the manuscript.



*Competing interests.* The authors declare that they have no conflict of interest.

*Acknowledgements.* We thank Kevin Hodges for providing the cyclone tracking code TRACK, and Helen Dacre for providing an initial version of the cyclone composite code. The authors wish to acknowledge CSC – IT Center for Science, Finland, for computational resources and ECMWF for providing the ERA5 reanalysis which is available from the Copernicus Climate Change Service Climate Data Store. This work was supported by MONITUHO project, the ERA4CS WINDSURFER project, and Academy of Finland Flagship funding (grant numbers 337549 and 337552).

520




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
