# Peer review of "Characteristics of extratropical cyclones and precursors to windstorms in Northern Europe"

_Weather and Climate Dynamics, 2021_

## Author Comment (AC1)

**Response to Reviewer 1 - "Characteristics of extratropical cyclones and precursors to windstorms in Northern Europe"**

Terhi K. Laurila, Hilppa Gregow, Joona Cornér and Victoria A. Sinclair

September 28, 2021

We thank the reviewer for constructive comments on our submitted manuscript. We have addressed all specific points raised by the reviewer (copied here and shown in black text), and include our response (in blue) below.

**Reviewer 1**

General Comments

The paper uses the ERA5 reanalysis dataset to obtain 40 years worth of Lagrangian meteorological fields for extratropical cyclones crossing into Northern Europe. Since the paper uses a new reanalysis data set to examine extratropical cyclones and their precursors over Northern Europe, it is novel. I find the composite results interesting and useful not only for specialists in the field, but also for a more general audience.

In the paper, composite analysis is used to characterize and understand Northern European extratropical cyclones, while ensemble sensitivity analysis is used to find correlations between precursor fields and the selected response functions: Minimum Mean Sea-Level Pressure (MSLP) and 10m wind gusts. Four precursor fields were selected: 850 hPa potential temperature, total column water vapor, and 300 hPa wind speed and potential vorticity. Regressions were computed for 1, 2, and 3 days before extrema in MSLP and 10m wind gusts.

The results are presented in a series of 18 figures that are legible, well-labeled, and generally of a high quality. Some notable results in the context of cyclones affecting Northern Europe include; no significant trends in cyclones were found over the 1980-2019 period; summer cyclones are less intense than winter ones; 1995-2005 lull in cyclone activity, which can be attributed to anomalously lower numbers of windstorms over the period; windstorms are more common in the winter; overall, numbers of cyclones do not show an appreciable seasonal cycle over Northern Europe; windstorms have distinct genesis regions compared to non-windstorm cyclones; windstorms preferentially genesis over the sea

while non-windstorms that affect Northern Europe genesis over Northern Europe; summer tracks are further poleward than winter tracks, and summer cyclones live longer.

The paper reads well, and it also establishes the scientific relevance of the study well. The paper also embeds its results in the established literature. I can recommend this paper for publication after some major revisions are made.

Major Comments

While the nature and characteristics of extratropical cyclones are of interest to practitioners and specialists in the field, I found the ensemble sensitivity part of the paper quite weak. The paper makes causal inferences when the method as is does not support causal inferences. The causal inferences from the Dacre et al., 2019 study were valid because the authors isolated the precursor fields from the effects of the cyclone itself before the regression. In the present study, all that can be said is that stronger baroclinic and moisture conditions in a developing cyclone are associated with stronger maximum cyclones. When put like that, the results of the ensemble sensitivity analysis don't seem as exciting. Thus all causal inferences (like L328 or L339-L342) should be removed prior to publication. Changing "leads to" or "causes" to "is associated with" may address this problem, but a broader rethink of how the method is used could lead to a far superior manuscript.

Thank you for a good comment. We agree that since the ensemble sensitivity analysis uses correlation we may not imply the results as causalities but rather in a way that the response function and the precursor are associated with each other. Although the influence of the cyclone is removed in the study by Dacre et al. (2019), there are other papers which have used the ensemble sensitivity analysis that have not removed the cyclones, e.g. Dacre and Gray (2013) and Garcies and Homer (2009). In addition, the sensitivity results without cyclone removal are more useful for forecasters since they look at the full fields of for example total column water vapour, potential vorticity etc. and rarely just at the background state. In our study, we think that finding correlations between the response functions and precursors is a valuable contribution to the literature. Therefore, we do not apply the cyclone removal to our current paper but instead, the sentences with causation interpretations are revised as suggested: to change "leads to" or "causes" to "is associated with".

References:

Dacre H.F., O. Martínez-Alvarado, C.O. Mbengue, 2019: Linking atmospheric rivers and warm conveyor belt airflows. J. Hydrometeorol., 20, pp. 1183-1196, https://doi.org/10.1175/JHM-D-18-0175.1.

Dacre H.F., and S.L. Gray, 2013: Quantifying the climatological relationship between extratropical cyclone intensity and atmospheric precursors. Geophys. Res. Lett., 40, 2322– 2327, doi:10.1002/grl.50105.

Garcies L. and V. Homar, 2009: Ensemble sensitivities of the real atmosphere: application to Mediterranean intense cyclones. Tellus A, 61: 394-406. https://doi.org/10.1111/j.1600-0870.2009.00392.x

Minor Comments

Figure 2 a,c,e: Make the origin non-zero so that the bars aren't unusually long and so that the variation among bars, not length of bars, is emphasized.

Figure 2 is now modified based on the comments from both reviewers: we now include only absolute numbers (i.e. the left panel), the origin is set to non-zero to see the variation among bars better, and we added horizontal lines to show the annual means.

Figure 3: Label the y-axis in the figure.

The y-axis in Figure 3 is the number of cyclones, and the y-axis label is now added. In addition, we added the y-axis label to Figure 2 as well (shows also the number of cyclones).

L209: A citation is required for the Mann-Kendall reference.

Relevant references are now added.

L210-211: I would end the discussion at no statistically significant trend was found because this line reads as if the experimenter is beholden to a particular finding.

We decided to keep this text because the threshold for the p-value is somewhat subjective (e.g. 90th or 95th percentile) but we'd like to reassure the reviewer that we were not beholden to a finding that the trend was significant.

Is there any explanation for the drop in windstorms between the years 1995 and 2000?

This is an interesting question but really difficult/impossible to answer. As we see from Figure 2, there is a large inter-annual and decadal variability in the number of extratropical cyclones and windstorms. A similar result, a large inter-annual and decadal variability, was found for the mean wind speeds and the 98th percentile wind speeds in the North Atlantic and Europe (Laurila et al., 2021). Additionally, Laurila et al. (2021) showed that the inter-annual variations are regional and hence different parts of Europe can experience quite different wind conditions during the same year. Therefore, we would like to highlight here in this response that it is "normal" to the climate to have periods with a lower number of windstorms (and likewise periods with a higher number of windstorms).

Reference: Laurila, T.K., Sinclair, V.A., Gregow, H., 2021: Climatology, variability, and trends in near-surface wind speeds over the North Atlantic and Europe during 1979–2018 based on ERA5. Int J Climatol., 41: 2253– 2278. https://doi.org/10.1002/joc.6957

Section 6. It may be helpful to indicate the direction of cyclone propagation on figures 7 and 8 (left to right?).

The arrow showing the propagation direction of the cyclone is now added to Figures 7 and 8.

L322: This causal statement does not follow from the results presented.

The statement is revised to say "is associated with".

L325: This causal statement does not follow from the results presented.

The statement is revised to say "is associated with".

L326: This causal statement does not follow from the results presented.

The statement is revised to say "is associated with".

L329: This causal statement does not follow from the results presented.

The statement is revised to say "is associated with".

L337: This causal statement does not follow from the results presented.

The statement is revised to say "is associated with".

L357: This causal statement (and the others that follow) does not follow from the results presented.

These statements are revised to say "is associated with".

L430: less → fewer

Corrected.

L456: Sentence unclear

The previous sentence was: "In addition, the spatial patterns of MSLP and 850-hPa potential temperature in our composites are similar that are found in the composites of the 100 strongest extratropical cyclones in the Northern Hemisphere (Catto et al., 2010)." Now this sentence is revised to say: "In addition, the spatial patterns of MSLP and 850-hPa potential temperature in our composites are similar to the spatial patterns found by Catto et al., (2010) when they created composites of the 100 strongest extratropical cyclones in the Northern Hemisphere." We hope this revised sentence is clearer.

---

## Author Comment (AC2)

**Response to Reviewer 2 - "Characteristics of extratropical cyclones and precursors to windstorms in Northern Europe"**

Terhi K. Laurila, Hilppa Gregow, Joona Cornér and Victoria A. Sinclair

September 28, 2021

We thank the reviewer for constructive comments on our submitted manuscript. We have addressed all specific points raised by the reviewer (copied here and shown in black text), and include our response (in blue) below.

**Reviewer 2**

This paper investigates extratropical cyclones over Northern Europe, which are typically less studied in the literature, but have big impacts on the region. The cyclones are split into two groups – those that are classed as windstorms (i.e. having strong winds within the region) and those that are non-windstorms. The authors analyse the seasonality of these storms, their structure, lifecycle, and correlations with precursor fields (using ensemble sensitivity analysis). An interesting result is that the minimum MSLP of the windstorms are more strongly correlated with precursor fields than the maximum 10-m wind gusts. However, the strong link between the minimum MSLP and the wind gusts gives extra potential predictability.

The paper is very well written and enjoyable to read, with some minor grammatical issues (detailed below).

My two main criticisms are:

1. The way in which the "windstorms" are selected gives (as the authors mention) mostly the extratropical cyclones that occur over the sea, due to decreased friction. It would be valuable to know whether storms that have the biggest impacts (i.e. the strongest winds over land) have the same precursors. This would certainly be more relevant when considering how these features might change in the future. However, if the authors feel that this is outside the scope of the paper, a stronger justification of the selection of the storms would be good.

   Thank you for a good comment. We agree that from an impact point of view it would be valuable to consider windstorms as extratropical cyclones that have strong wind gusts over land. Even

over land, using just one threshold would not be suitable since the wind speeds over land on the west coast of Norway are much stronger than the wind speeds over land over Finland, and we would most likely end up with just windstorms on the Norwegian coast. Therefore, we would need to define windstorms over land using a geographically varying wind speed threshold which is very complicated to do in conjunction with individual cyclone tracks which move through different geographical areas. In our paper, we focus more on the dynamics and the structure of windstorms rather than impacts. Therefore, we think that our choice of selecting windstorms based on the 90th percentile of wind gusts over the whole Northern Europe box is valid. We have now justified this choice better in the paper by adding a sentence in lines 141–142 that says: "Furthermore, since our study focuses more on the dynamics and the structure of windstorms rather than impacts, the threshold we use considers wind gusts over the whole Northern Europe box rather than only wind gusts over land.".

2. The ensemble sensitivity analysis is based on correlations between fields, and does not necessarily imply causation. Many of the phrases "leads to" or "causes" (e.g. line 185), should be changed to reflect this.

   Thank you for a good comment. We agree that since the ensemble sensitivity analysis uses correlation we may not imply the results as causalities but rather in a way that the response function and the precursor are associated with each other. We have now revised the sentences with causation interpretations of "leads to" or "causes" to "is associated with".

Minor comments

1. Line 15: "than" → "as".
   Corrected.

2. Line 30: "and" → "with".
   Corrected.

3. Line 44: "to" → "of".
   Corrected.

4. Line 61: "of" → "for".
   Corrected.

5. Lines 51-66: The inclusion of some references would be useful here.
   We have added references to Faranda et al. (2017) and Krueger and Von Storch (2011) to this paragraph.

   References:

   Faranda, D., Messori, G. and Yiou, P., 2017: Dynamical proxies of North Atlantic predictability and extremes. Sci Rep 7, 41278, https://doi.org/10.1038/srep41278

   Krueger, O., and Von Storch, H. (2011). Evaluation of an air pressure–based proxy for storm activity. Journal of climate, 24(10), 2612-261, https://doi.org/10.1175/2011JCLI3913.1

6. Lines 79-80: This gives the impression that this study will also consider climate change. Perhaps it could be made clearer that this is a possibility for future study.

   We have shortly discussed about this in the discussion chapter (in lines 481–484) where we apply our sensitivity results to the known changes in the low-level temperature gradient and atmospheric moisture. However, we now changed the start of that sentence to say "future studies potentially could estimate" instead of "we can estimate".

7. Introduction: For some of the introduction, the terms "extratropical cyclones" and "windstorms" seem to be interchangeable until line 81. Then the difference is not made clear before the research questions. I suggest including a brief definition of extratropical cyclones versus windstorms.

   Thank you for the suggestion, we have now defined a windstorm in lines 62–63 where the word windstorm is mentioned for the first time.

8. Line 90: "How the spatial..." → "How does the spatial...".

   Corrected.

9. Line 111: Many other studies using TRACK use a truncation of T42 before identifying extratropical cyclones. Is this because you are using MSLP rather than vorticity? It would be useful to mention this difference.

   Yes, higher resolution is used for MSLP than for vorticity since the MSLP field is smoother than vorticity field. For example, Hodges et al. (2011) use T63 resolution for MSLP and T42 for vorticity for this reason. We have now added a following sentence to lines 115–116: "Many studies use T42 resolution for vorticity based tracking (e.g. Priestley et al., 2020), however, higher resolution of T63 is generally used for MSLP since its field is smoother than the vorticity field (Hodges et al., 2011)."

   References:

   Priestley M. D. K., Ackerley D., Catto J. L., Hodges K. I., McDonald R. E., and Lee R. W., 2020: An overview of the extratropical storm tracks in CMIP6 historical simulations. J. Clim., 33, 6315–43, https://doi.org/10.1175/JCLI-D-19-0928.1.

   Hodges, K. I., R. W. Lee, and L. Bengtsson, 2011: A comparison of extratropical cyclones in recent reanalyses ERA-Interim, NASA MERRA, NCEP CFSR, and JRA-25. J. Climate, 24, 4888–4906, https://doi.org/10.1175/2011JCLI4097.1.

10. Line 116: "with a 6..." → "within a 6...".

    Corrected.

11. Line 128: "differ from..." → "go from...".

    We corrected this to "vary from".

12. Line 138: "which" → "whose".

    Corrected.

13. Figure 2: I'm not sure you need both the left and right panels. Perhaps on the left panels you could just add a dashed line to indicate the mean.

    Figure 2 is now modified based on the comments from both reviewers: we now include only absolute numbers (i.e. the left panel), the origin is set to non-zero to see the variation among bars better, and we added horizontal lines to show the annual means.

14. Line 251: "less" → "fewer".

    Corrected.

15. Lines 313, 317, 379 and others: The use of the term "offset times" is confused in places. Line 317, for example, I would refer to the time of minimum MSLP as the "central" time (or similar). Then the offset is the difference from that central time.

    We have carefully revised the sentences throughout the paper so that the offset time refers to the difference from the time of the minimum MSLP / maximum gust and not to the 0 h timing.

16. Line 335 and throughout: "similar than" should be "similar to" everywhere it appears.

    This is now corrected throughout the manuscript.

17. Line 347: Why not just say "minimum MSLP" instead of "maximum intensity in terms of MSLP"?

    Good point. By writing it that way we wanted to highlight that we are investigating maximum intensity in two ways: in terms of MSLP and in terms of wind gusts. So both are maximum intensities although MSLP is minimum and gust is maximum. But we have now revised this sentence as suggested here.

18. Line 360: I think the sentence could end after "cold season".

    The sentence is revised as suggested.

19. Line 408: Perhaps the first sentence could be turned around. E.g. "The 10-m wind gusts are weakly sensitive to the 300-hPa PV".

    The sentence is revised as suggested.

---

## Author Response (AR2)

**Response to Reviewer - "Characteristics of extratropical cyclones and precursors to windstorms in Northern Europe"**

Terhi K. Laurila, Hilppa Gregow, Joona Cornér and Victoria A. Sinclair

October 25, 2021

Thank you for pointing out these minor technical issues. We have now made all of the recommended corrections except one. The reviewer suggested to change the wording "Sect." to "section" throughout the manuscript, however, we have kept the "Sect." since we believe this is the format required by the journal (https://www.weather-climate-dynamics.net/submission.html).